Corrected: Author correction

# Herbivorous turtle ants obtain essential nutrients from a conserved nitrogen-recycling gut microbiome

Yi Hu [1], Jon G. Sanders [2,3], Piotr Łukasik [1], Catherine L. D'Amelio[1], John S. Millar [4], David R. Vann [5], Yemin Lan [6], Justin A. Newton[1], Mark Schotanus[7], Daniel J.C. Kronauer[8], Naomi E. Pierce [2], Corrie S. Moreau [9], John T. Wertz [7], Philipp Engel [10] & Jacob A. Russell [1]

Nitrogen acquisition is a major challenge for herbivorous animals, and the repeated origins of herbivory across the ants have raised expectations that nutritional symbionts have shaped their diversification. Direct evidence for N provisioning by internally housed symbionts is rare in animals; among the ants, it has been documented for just one lineage. In this study we dissect functional contributions by bacteria from a conserved, multi-partite gut symbiosis in herbivorous *Cephalotes* ants through in vivo experiments, metagenomics, and in vitro assays. Gut bacteria recycle urea, and likely uric acid, using recycled N to synthesize essential amino acids that are acquired by hosts in substantial quantities. Specialized core symbionts of 17 studied *Cephalotes* species encode the pathways directing these activities, and several recycle N in vitro. These findings point to a highly efficient N economy, and a nutritional mutualism preserved for millions of years through the derived behaviors and gut anatomy of *Cephalotes* ants.

[1] Department of Biology, Drexel University, Philadelphia, PA 19104, USA. [2] Department of Organismic and Evolutionary Biology, Harvard University, Cambridge, MA 02138, USA. [3] Department of Pediatrics, The University of California San Diego, La Jolla, CA 92093, USA. [4] Department of Medicine, Institute of Diabetes, Obesity and Metabolism, University of Pennsylvania, Philadelphia, PA 19104, USA. [5] Department of Earth and Environmental Science, University of Pennsylvania, Philadelphia, PA 19104, USA. [6] School of Biomedical Engineering, Science and Health systems, Drexel University, Philadelphia, PA 19104, USA. [7] Department of Biology, Calvin College, Grand Rapids, MI 49546, USA. [8] Laboratory of Social Evolution and Behavior, The Rockefeller University, New York, NY 10065, USA. [9] Department of Science and Education, Field Museum of Natural History, Chicago, IL 60605, USA. [10] Department of Fundamental Microbiology, University of Lausanne, 1015 Lausanne, Switzerland. These authors contributed equally: Yi Hu, Jon G. Sanders. Correspondence and requests for materials should be addressed to Y.H. (email: yh332@drexel.edu)

Nitrogen (N) is a key component of living cells and a major constituent of the nucleic acids and proteins directing their structure and function. Like primary producers[1], herbivorous animals face the challenge of obtaining sufficient N in a world with limited accessible forms of this element, suffering specifically due to the low N content of their preferred foods[2]. The prevalence of herbivory is, hence, a testament to the many adaptations for sufficient N acquisition. Occasionally featured within these adaptive repertoires are internally housed, symbiotic microbes. Insects provide several examples of such symbioses, with disparate herbivore taxa co-opting symbiont N metabolism for their own benefit[3–5]. Such tactics are not employed by all insect herbivores[6], however, and few studies have quantified symbiont contributions to host N budgets[7–10].

Ants comprise a diverse insect group with a broad suite of diets. Typically viewed as predators or omnivores, several ants are functional herbivores, with isotopic N ratios overlapping those of known herbivorous insects[11,12]. While occasionally obtaining N from tended sap-feeding insects, most herbivorous ants are considered plant canopy foragers, scavenging for foods such as extrafloral nectar, pollen, fungi, vertebrate waste, and plant wound secretions[12]. Quantities of usable and essential N in such foods are limiting[13,14]. Hence, the repeated origins of functional herbivory provide a useful natural experiment, enabling tests for symbiotic correlates of N-limited diets. The relatively high prevalence of specialized bacteria within herbivorous ant taxa suggests such a correlation[15,16]. However, N provisioning by internally housed symbionts has only been documented for carpenter ants (tribe Camponotini), whose intracellular *Blochmannia* provide them with amino acids made from recycled N[17].

Herbivorous cephalotines (i.e., sister genera *Cephalotes* and *Procryptocerus*) and ants from other herbivore genera (e.g., *Tetraponera* and *Dolichoderus*) exhibit hallmarks of a symbiotic syndrome distinct from that in the Camponotini. Large, modified guts with prodigious quantities of extracellular gut bacteria make up one defining feature[18–21]. Also characteristic are the oral–anal trophallaxis behaviors transmitting symbionts between siblings[18,22–24] and the domination of gut communities by host-specific bacteria[15,25,26]. Such symbiotic "hotspots" stand out in relation to several ant taxa, which show comparatively low investment in symbiosis[16,20].

N provisioning by bacterial symbionts in these ants has been hypothesized as a mechanism for their success in a seemingly marginal dietary niche[12,15]. To investigate this, we focus on turtle ants, i.e., the genus *Cephalotes* (Fig. 1). With ~115 described species[27], stable isotopes place these arboreal ants low on the food chain[12,28]. Workers are canopy foragers, consuming extrafloral nectar and insect honeydew, fungi, pollen, and leaf exudates[29,30]. *Cephalotes* also consume mammalian urine and bird feces, which both contain large quantities of waste N, accessible only through the aid of microbes. The remarkably conserved gut microbiomes of cephalotines[25,31] have been proposed as an adaptation for these N-poor and N-inaccessible diets. Here we measure symbiont N provisioning in *Cephalotes varians* and characterize gene content within the gut microbiomes of 17 *Cephalotes* species (Supplementary Table 1), describing symbiont N metabolism across 46 million years of evolutionary history.

Our results reveal that core gut bacteria from *Cephalotes* recycle common nitrogenous wastes, including urea and uric acid. Essential amino acids containing recycled N are synthesized by these microbes and obtained by host workers in substantial quantities. Nearly all core symbionts can make the majority of essential and non-essential amino acids, but just a subset encode particular pathways to recycle N waste. Recyclers involved in uric acid and urea breakdown hail from lineages in the Burkholderiales, Opitutales, and Rhizobiales, and (meta)genome annotations suggest that either one of the latter two taxa is required to process urea produced by Burkholderiales through purine, xanthine, or uric acid metabolism. Metabolic assessments of cultured core symbionts, along with their phylogenetic placement, suggest long-standing conservation of this complementary N-recycling partnership. Further analysis of arginine metabolism reveals a broader capacity of gut symbionts to synthesize urea, revealing hallmarks of a highly efficient N economy in these N-limited, arboreal ants.

## Results

**No evidence for symbiont N fixation.** Atmospheric N fixation is executed by bacterial symbionts of some invertebrates[32–34], and prior detection of nitrogenase genes in ants[15,26] has led to the proposal that symbiotic bacteria fix nitrogen for these hosts. To test this, three *C. varians* colonies were subjected to acetylene reduction assays within hours of field capture. In three separate experiments, no detectable ethylene was produced after 16 h of ant incubation in the presence of acetylene (Supplementary Table 2). This finding resembled that from a prior experiment on lab-reared *C. varians*, which utilized smaller numbers of workers[15]. Collectively, both argue against active and substantial N fixation by *Cephalotes* symbionts.

**Ants acquire small amounts of N from symbiont-upgraded dietary AAs.** Based on precedents from intracellular symbionts of insects[10,17,35], we next tested whether gut bacteria could upgrade ant diets, transforming non-essential or inaccessible N compounds into essential amino acids that are acquired by hosts. Our efforts first focused on glutamate, an important precursor in the synthesis of many amino acids. *C. varians* workers from three colonies were reared on artificial diets varying in the presence of antibiotics and the presence of heavy isotope-labeled glutamate containing $^{15}N$ in our first experiment and $^{13}C$ in our second trial. Ingredients in the artificial diet consisted of vitamins, minerals, salts, growth factors, carbohydrates, and amino acids. The original recipe[36] was modified so that only non-essential amino acids were included (tyrosine, aspartate, asparagine, serine, alanine, cysteine, glycine, glutamate, glutamine, and proline, but not arginine), each at a 200 mg/ml concentration.

Heavy isotope enrichment in the free amino acid pools from worker hemolymph was assessed via gas chromatography–mass spectrometry (GC–MS) (Supplementary Table 3). This allowed us to quantify the use of C or N from glutamate to synthesize host-associated amino acids. Diverse metabolic pathways enable the use of glutamate-derived C or N. Some are encoded by animals and bacteria alike, such as those for non-essential amino acid biosynthesis, and transamination reactions, where the $NH_3$ group from glutamate is donated to the carbon skeletons of amino acid precursors (Supplementary Fig. 1). In spite of this complexity, symbiont contributions to host amino acid pools, via dietary glutamate, would be partially evidenced if elevated $^{13}C$ or $^{15}N$ signal within ant hemolymph (i.e., on heavy glutamate diets) were eliminated through antibiotic treatment. Results from the $^{13}C$ experiment would be most telling in many ways. For example, after diversion through the tricarboxylic acid cycle, glutamate-derived carbon could be used by bacteria to synthesize the isoleucine, lysine, methionine, and threonine. By definition, these essential amino acids are nutrients that animals cannot synthesize de novo. Hence, any elevation in their $^{13}C$ signal on the heavy glutamate diet, coupled with a lack of such enrichment under antibiotic treatment, would directly implicate symbionts in contributing to host nutrition through the use of dietary glutamate.

Antibiotic treatment successfully suppressed gut microbial load in this and all below experiments, leaving behind communities that were drastically altered in composition (Supplementary

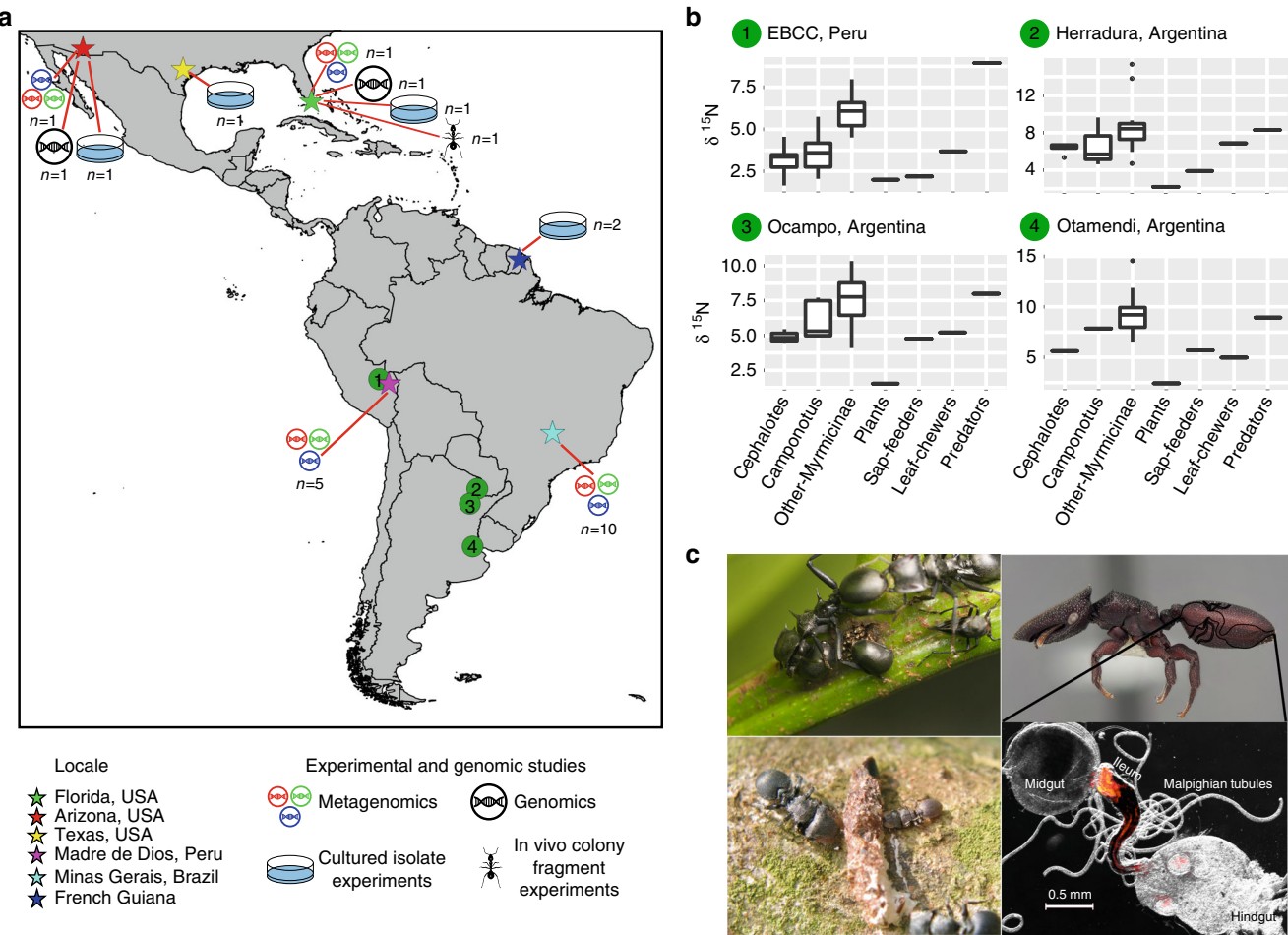

**Fig. 1** Ecology of *Cephalotes* ants and origins of specimens used in our study. **a** Map showing sampling locales for cephalotines (*Cephalotes* and *Procryptocerus*) used in this study (stars), the activities they were used for, along with sample size (i.e., number of species). Numbered circles show sites of ant sampling in two prior studies[12,28], from which stable nitrogen isotope data were extracted and plotted here. The map was constructed using an open source geographic information system QGIS software version 2.18. **b** Nitrogen isotope data from prior studies[14,31] targeting several New World locales. Each separate box plot graphs the median (center line), the interquartile range (i.e., from the top to the bottom of each box), values equal to 1.5 times the interquartile range (tops and bottoms of whiskers), along with outliers. Graphed are isotope ratios for *Cephalotes* ants, other ants in their subfamily (Myrmicinae), and *Camponotus* ants, which host known N-recycling bacteria. Also plotted are data for plants, insect herbivores, and insect carnivores from the same locales. Isotope fractionation results in a gradual increase in the relative amounts of heavy nitrogen ($^{15}$N), as one moves up the food chain. As such, ants with low amounts of the heavy nitrogen isotope, shown here using the δ$^{15}$N statistic, are argued to feed at low trophic levels. **c** *C. atratus* workers tend honeydew-producing, ant-mimicking membracids (upper left, image credit: Jon Sanders). *C. eduarduli* and *C. maculatus* (smaller worker) feeding on bird droppings (lower left, from Fig. 2b of Russell et al.[16], used with permission from Myrmecological News and photographer Scott Powell). Soldier caste of *C. varians* with an outlined digestive tract (upper right, image credit: Corrie Moreau). A FISH microscopy image of a digestive tract from a *Cephalotes* worker is shown at lower right (image credit: Piotr Lukasik). Note the large bacterial mass in the ileum near the midgut–ileum junction, the site where N wastes are emptied via Malpighian tubules

Fig. 2). In addition, antibiotic-treated workers in this and the below experiments survived treatments at rates sufficient for subsequent data generation (Supplementary Fig. 3). GC–MS analyses of ant hemolymph revealed that workers absorbed nutrients from the administered diets, as hemolymph glutamate pools showed ~5–6% enrichment for heavy isotopes on the heavy vs. light isotope diets in the absence of antibiotics ($p = 0.0033$ $^{15}$N vs. $^{14}$N diet, one-way analysis of variance (ANOVA) with post hoc Tukey's HSD (Honestly Significant Difference) test, Supplementary Table 3; $p = 0.0018$ $^{13}$C vs. $^{12}$C diet, Kruskal–Wallis test with post hoc Mann–Whitney test). Yet, symbiont suppression had only a small impact on ants' acquisition of C and N from dietary glutamate. For instance, on the $^{13}$C-glutamate diet, antibiotic treatment reduced the fraction of heavy isotope-bearing isoleucine ($p = 0.0147$, one-way ANOVA with post hoc Tukey's HSD test), leucine ($p = 0.0004$, one-way ANOVA with

post hoc Tukey's HSD test), threonine ($p = 0.0029$, Kruskal–Wallis test with post hoc Mann–Whitney test), and tyrosine ($p = 0.0169$, Kruskal–Wallis test with post hoc Mann–Whitney test) in worker hemolymph (Supplementary Fig. 4), relative to estimates on this same diet without antibiotics. But effect sizes for each amino acid were small, with changes of just 1.3–2.6% in ants with suppressed microbiota. On the $^{15}$N-glutamate diet (Supplementary Fig. 5), only phenylalanine ($p = 0.045$) showed heavy isotope enrichment in untreated vs. antibiotic-treated workers, again with a small effect size (2.9%). To summarize, findings that four host-associated essential amino acids (isoleucine, leucine, phenylalanine, and threonine) were impacted by symbiont presence suggests some role for the gut microbiome in upgrading non-essential amino acids from the diet. However, small effect sizes raise questions on the importance of such activities to overall host N budgets.

**Ants acquire large quantities of recycled N from symbionts**. In nature, *Cephalotes* ants consume N-rich vertebrate excreta, including bird droppings and mammalian urine. Uric acid and urea are the predominant forms of N waste in these food sources. These compounds are also thought to be major components of the physiological N waste delivered to the gut by the insects' Malpighian tubules. In turtle ants, Malpighian tubules empty into the ileum, which houses large masses of extracellular bacteria (Fig. 1) comprising the majority of known core symbiont taxa[22]. Since insects generally lack the capacity to convert dominant nitrogenous waste compounds, like uric acid or urea, into usable forms of N, it has been posited for *Cephalotes* and other ant herbivores that extracellular gut symbionts recycle such N waste, converting recycled N into essential amino acids that are acquired by hosts[18].

To test this hypothesis, we implemented an experiment similar to those described above, supplementing ant diets with $^{15}$N isotope-labeled urea instead of glutamate. Unlike the aforementioned experiments, however, urea was the only source of N in the present trials (see Supplementary Methods). After consuming diets with heavy urea, 15 of the 16 detectable amino acids in *C. varians* hemolymph were enriched for the heavy isotope signal when compared to diets with light ($^{14}$N) urea (i.e., all but asparagine; significant *p*-value range: 1.92E−14 to 0.0210, one-way ANOVA with post hoc Tukey's HSD test for normally distributed data after logit transformation or Kruskal–Wallis test with post hoc Mann–Whitney test for non-normal data after logit transformation; Fig. 2; Supplementary Table 3). On the $^{15}$N diet, antibiotic treatment strongly reduced the heavy isotope signal in these same 15 amino acids (significant *p*-values ranged from 1.92E−14 to 0.0410), directly implicating bacteria in the use of diet-derived N waste. The impact of bacterial metabolism on worker N budgets was substantial, with 15–36% enrichment for heavy essential amino acids in hemolymph of symbiotic, versus aposymbiotic, ants within 5 weeks on the experimental diet.

**Metagenomic analyses show strong taxonomic conservation**. To address the mechanisms behind symbiont N recycling and upgrading, we used shotgun Illumina HiSeq sequencing to characterize microbiomes colonizing the midgut and ileum. Eighteen sequence libraries were generated across 17 *Cephalotes* species collected from 4 locales (Fig. 1; Supplementary Table 1). Two of these libraries came from our experimental model species, *C. varians*. Shotgun sequencing efforts yielded median values of 32,706,498 reads and 143.625 Mbp of assembled scaffolds per library (Supplementary Table 4). The median N50 for scaffold length was 1106.5 bp.

A prior study suggested that >95% of the *Cephalotes* gut community is comprised of core symbionts from host-specific clades, i.e., lineages of bacteria distributed across *Cephalotes* and its sister genus, *Procryptocerus*, that are, thus far, found only in these hosts[37]. To assess whether the dominant bacteria sampled here came from such specialized groups we extracted 16S ribosomal RNA (rRNA) fragments >200 bp from each metagenome library. Top BLASTn hits identified in the National Center for Biotechnology Information (NCBI) non-redundant nucleotide database were downloaded for each sequence, and jointly used in a maximum likelihood phylogenetic analysis. In the resulting tree (Supplementary Fig. 6), 94.4% of our 335 *Cephalotes* symbiont sequences grouped within 10 cephalotine-specific clades that included sequences from prior studies. Inferences on metagenome content have, hence, been made using partial genomes from the dominant, specialized core taxa.

Classification of assembled scaffolds took place using USEARCH comparisons against public reference genomes in IMG (Integrated Microbial Genomes) and the KEGG (Kyoto

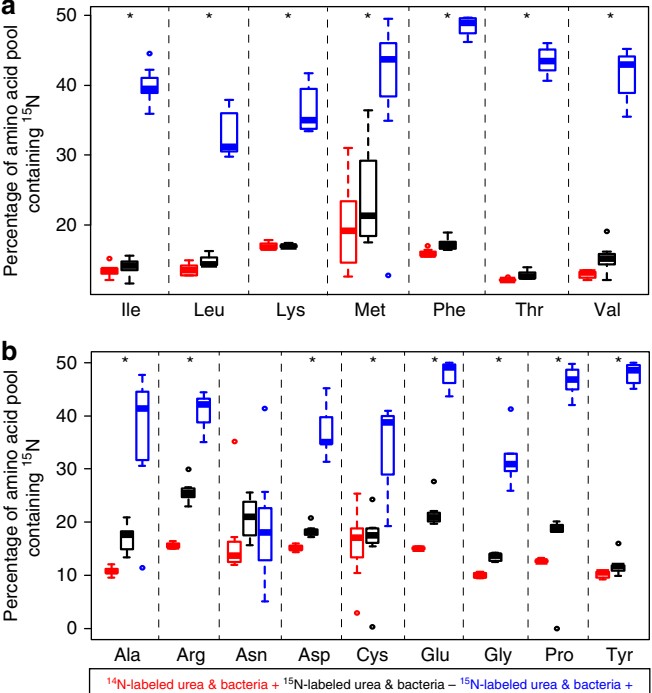

**Fig. 2** Symbiont suppression reduces proportions of $^{15}$N-labeled amino acids in hemolymph of *Cephalotes varians* workers consuming $^{15}$N-labeled urea. **a** Essential and **b** non-essential amino acids in ant hemolymph measured through GC–MS. Asterisks indicate that $^{15}$N in essential amino acids from ants consuming $^{15}$N-labeled urea (blue) was significantly higher than that in antibiotic-treated ants on this same diet (black) and in those consuming diets with unlabeled urea (red). Details on sample sizes for each treatment can be found in Supplementary Data 6. Note that each separate box plot graphs the median (center line), the interquartile range (i.e., from the top to the bottom of each box), values equal to 1.5 times the interquartile range (tops and bottoms of whiskers), and outliers. Amino acids for which we observed significantly higher heavy isotope signal in the no antibiotic (blue) versus antibiotic (black) treatments, on the heavy isotope diet, are indicated with an asterisk. Statistics are described in the Supplementary Methods and Supplementary Table 3. In short, asterisks reveal results from Tukey's post hoc tests (for normally distributed data after logit transformation) or Wilcoxon rank sum tests (for non-normal data after logit transformation)

Encyclopedia of Genes and Genomes) database. Results from these analyses paralleled the aforementioned 16S rRNA-based discoveries of a highly conserved core microbiome, in which different *Cephalotes* species host microbes from a limited set of lineages found exclusively across this genus and the *Procryptocerus* sister genus (Supplementary Fig. 6). These symbiont lineages nest within described orders from the Proteobacteria, Verrucomicrobia, and Bacteroidetes. While most symbionts await formal taxonomic descriptions, work on two core lineages shows that some comprise novel genera[38] and, in at least one case, a novel family[39].

In all metagenomes, *Cephaloticoccus* symbionts[38] from the order Opitutales were the most dominant, with scaffolds from these bacteria typically forming a single "cloud" differentiated from others by depth of coverage and %GC content (Fig. 3). Xanthomonadales scaffolds were ubiquitous and typically abundant, with multiple scaffold clouds often evidencing co-existence of distinct lineages, with up to ~10% divergence in average %GC content. Less abundant, though still ubiquitous across metagenomes, were clouds of scaffolds from the Pseudomonadales,

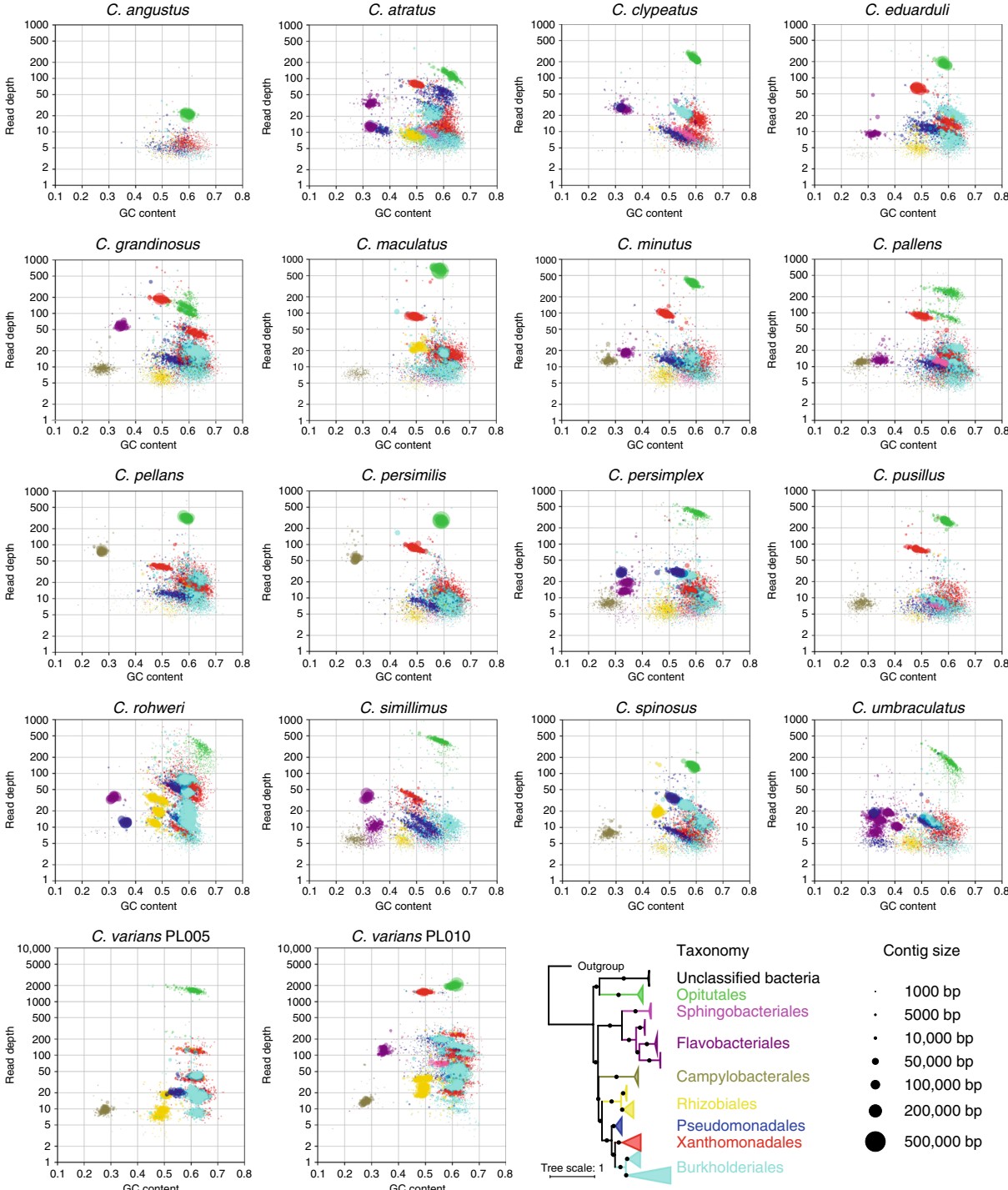

**Fig. 3** Taxon-annotated GC-coverage plots for 18 *Cephalotes* metagenomes reveal strong microbiome conservation. Assembled scaffolds in each metagenome are plotted based on their %GC content (*x*-axis) and their depth of sequencing coverage (*y*-axis, log scale). Bacterial genomes vary in %GC genome content and core symbionts show variable abundance; these plots, thus, illustrate the co-existence of numerous dominant symbiont strains in *Cephalotes* worker guts. The phylogeny at lower right, based on 16S rRNA sequences from our metagenomes, identifies the *Cephalotes*-specific clades from which nearly all of our sequence data have been obtained. Colors on the phylogeny match those in the blob plots, illustrating the taxa to which scaffolds were assigned. Circle size indicates length of each scaffold. Not shown here are scaffolds binning to Hymenoptera or to unclassified organisms. Note that for each metagenome library, DNA was obtained from a total of 10 workers from a single colony

Burkholderiales, and Rhizobiales. Multiple scaffold clouds (e.g., at differing depths of coverage) from each of these orders were found in the same metagenomes, suggesting the co-existence of multiple, related strains within most microbiomes. Unlike these core groups, Flavobacteriales, Sphingobacteriales, and

Campylobacterales were common but not ubiquitous, echoing previous results using 16S rRNA amplicon sequencing[31,37]. The absence or rarity of these symbionts in several metagenomes is also reflected in the presence/absence calls for N-metabolism genes in these taxa (Supplementary Data 1).

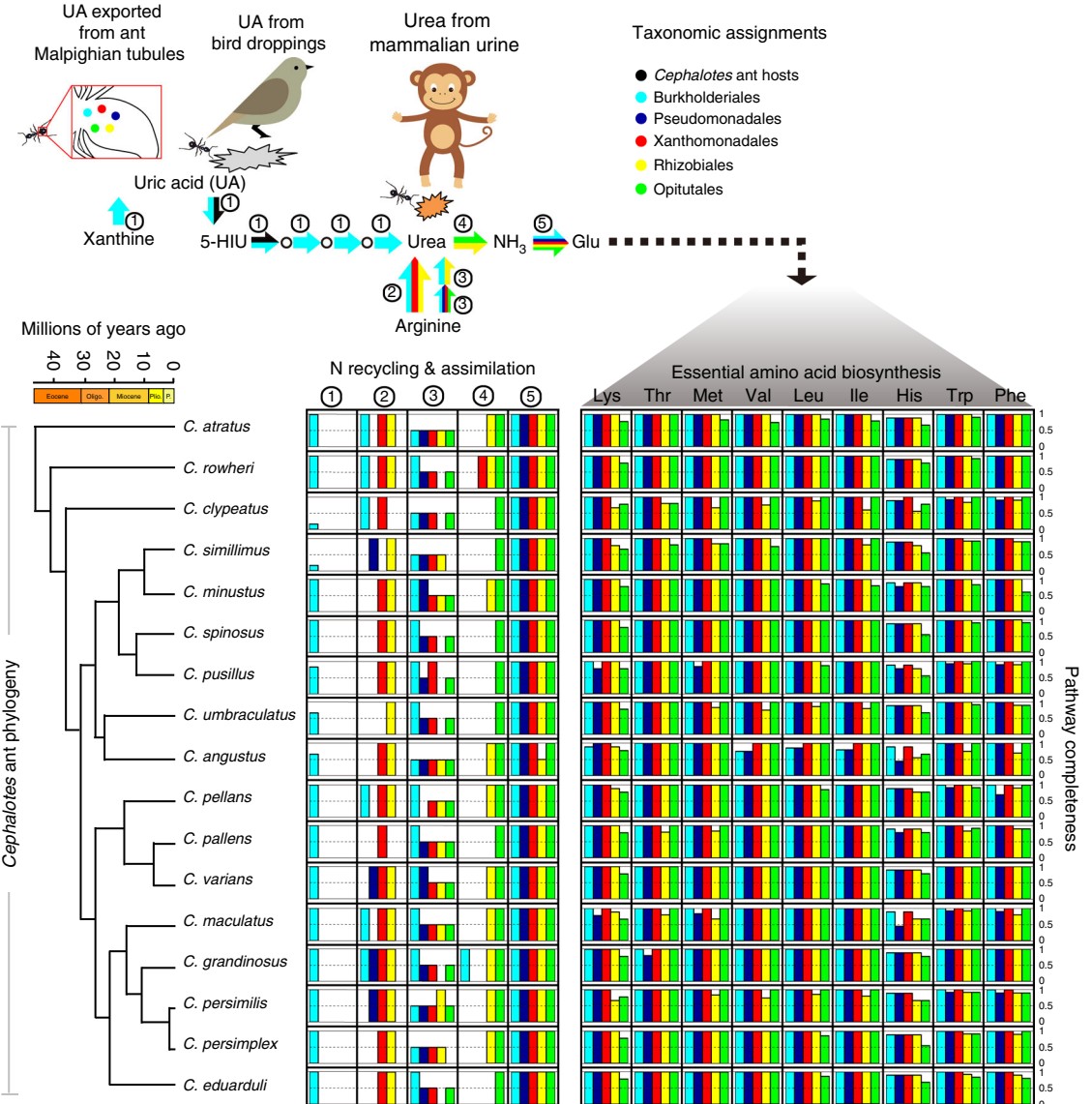

**Fig. 4** Pathways for N-waste recycling and amino acid biosynthesis and their distributions across core gut symbionts from 17 *Cephalotes* species. Various symbiotic gut bacteria convert N wastes into ammonia, incorporate ammonia into glutamate, and synthesize essential amino acids. The proposed model in the upper panel shows sources of the N-waste uric acid (bacterial metabolism via xanthine degradation, bird droppings, host ant waste metabolism via Malpighian tubule delivery) and urea (mammalian urine, uric acid metabolism, and arginine metabolism). Arrows in this panel are colored to reflect taxonomy of the core *Cephalotes*-specific microbes participating in these steps in multiple metagenomes. Numbers near arrows link particular pathways to bar graphs (below), which in turn plot pathway completeness (i.e., proportion of all genes present) for the dominant core taxa in each metagenome. At left on the lower panel is the phylogeny of *Cephalotes* species used for metagenomics including a chronogram dating divergence events in these species' history[27]. The bird and monkey images used in the upper panel were created using Adobe Illustrator CS6 (v. 16.0.0) and the ant image is an Adobe Illustrator clipart image

**N fixation genes appear absent in metagenomes**. Consistent with our acetylene reduction experiments using *C. varians*, IMG/ER-based annotation recovered no N-fixation genes in any of the 18 metagenome libraries. This absence encompassed genes encoding the molybdenum-containing nitrogenase system (i.e., *nifD*, *nifH*, *nifK*), and those from the iron-only (*anfD*, *anfG*, *anfH*, *anf*K) and vanadium-containing (*vnfD*, *vnfG*, *vnfH*, *vnfK*) systems (Supplementary Data 1). To complement these efforts, we also used previously sampled *nifH* sequences from *Cephalotes* workers as query sequences[15] in BLASTn and tBLASTx searches against our best-sampled metagenome, from *C. varians* colony PL010. No significant hits were obtained in these searches. Together, our experiments and metagenomics suggest that prior observations of *nifH* genes in *Cephalotes* workers arose from

detection of rare or contaminant bacteria[15] or from bacteria colonizing a portion of the gut not targeted in our metagenomic study.

**Urease is ubiquitous but limited to a subset of core symbionts**. Matching our discovery of symbiotic N recycling of dietary urea in *C. varians* were findings of *ureA*, *ureB*, and *ureC* genes in both *C. varians* metagenomes and in those of the 16 additional *Cephalotes* species (Fig. 4; Supplementary Figs. 7, 8; Supplementary Data 1). The presence of complete gene sets for the core protein subunits of the urease enzyme in all sampled microbiomes suggests that symbionts from most, if not all, *Cephalotes* species can make ammonia from N-waste urea. Taxonomic classification for urease gene-encoding scaffolds suggested that

abundant *Cephaloticoccus* symbionts (order: Opitutales) encoded all three core urease genes. Complete copies of each gene were found on a single Opitutales-assigned scaffold within 15 of 18 metagenomes. Genes encoding the urease accessory proteins (*ureF*, *ureG*, and *ureH*) were often found on these same scaffolds, with a strong trend of conserved architecture for this gene cluster (Supplementary Fig. 7).

Urease genes were occasionally assigned to other bacteria (Fig. 4; Supplementary Data 1), suggesting that more than one symbiont can participate in this recycling function. Notable were cases from *C. rohweri* (Xanthomonadales), *C. grandinosus* (Burkholderiales), and *C. eduarduli* (unclassified Bacteria), which hosted additional bacteria encoding the core and accessory proteins required for urease function. In these cases, urease genes mapped to single scaffolds with identical gene order to that seen for *Cephaloticoccus* (Supplementary Fig. 7). Urease function was also inferred for Rhizobiales bacteria in several *Cephalotes* species. Rhizobiales-assigned scaffolds encoding urease genes differed from those of *Cephaloticoccus* with respect to gene order, the presence of the *ureJ* accessory gene, and the existence of a gene fusion between *ureA* and *ureB* (Supplementary Fig. 7).

A maximum likelihood phylogenetic analysis of UreC proteins encoded by the sampled microbiomes identified two distinct *Cephalotes*-specific lineages (Supplementary Fig. 9). The first (bootstrap support = 99%) consisted of Rhizobiales-assigned UreC proteins, with relatedness to homologs from various families in the Rhizobiales. The second (bootstrap support = 93%) comprised of proteins assigned to Opitutales, Burkholderiales, Xanthomonadales, and unclassified Bacteria, showing relatedness to homologs from bacteria in the Rhodocyclales (Betaproteobacteria).

**Most core symbionts assimilate ammonia and synthesize AAs.** The above results provide genetic mechanisms to explain symbiont-mediated urea recycling in *C. varians*, suggesting a broad distribution for this function across the genus *Cephalotes*. Also necessary to explain our experiments are: (1) symbiont genes to assimilate the ammonia made from urease-driven urea degradation, and (2) symbiont genes using this assimilated N to synthesize amino acids.

Metagenome-derived predictions of functional capacity met these expectations in *C. varians* and all 16 other turtle ants. However, in contrast to our findings of one to three urea recyclers per host, genes encoding ammonia assimilation and amino acid biosynthesis were assigned to all core symbiont taxa, suggesting extensive metabolic redundancy. Within all metagenomes, numerous taxa encoded complete gene sets for ammonia assimilation (e.g., Fig. 4; Supplementary Figs. 1, 8, 10; Supplementary Data 1). Similarly, complete gene sets for the synthesis of each essential and non-essential amino acid were found in bacteria from all metagenomes, though a small number of genes were missing from certain taxonomic bins (Fig. 4; Supplementary Figs. 1, 8; Supplementary Data 1). Some of these missing genes may have stemmed from low sequencing coverage, as in *C. angustus*. The metagenome from this host had the lowest depths of microbial sequencing coverage in our dataset (Fig. 2), and also contained Xanthomonadales and Burkholderiales bins with an abnormally large number of "missing" genes in several biosynthetic pathways (Fig. 4; Supplementary Fig. 8). The absence of genes could also stem from limits to current pathway annotations, as non-homologous enzymes can converge upon the same functions[40]. It could further arise due to metabolite exchange among hosts or microbes[41]. Either of these latter two explanations could explain why a single biosynthetic step seemed consistently absent for methionine biosynthesis in Rhizobiales

and Pseudomonadales bins, or why the nearly complete histidine biosynthesis pathways recovered in Xanthomonadales and Burkholderiales bins were consistently "missing" the same step (Fig. 4; Supplementary Fig. 8).

**Uric acid and other precursors of N-waste urea.** Outside of the *Cephalotes* diet (e.g., animal urine), urea can be derived from a small variety of waste products, including uric acid. This compound is thought to be a major component of ants' physiologically derived N waste, and is a suspected component of *Cephalotes* ant diets given their natural attraction to bird excreta. To analyze symbiont capacities to recycle N from uric acid we examined gene content in the pathway converting this compound into urea. As seen for urease function, metagenomic analyses implicate just a subset of the microbiota in uric acid recycling, with uric acid pathway genes assigning to Burkholderiales symbionts in nearly all metagenomes (Figs. 3 and 4; Supplementary Figs. 7, 8; Supplementary Data 1). Importantly, the *puuD* uricase homolog was detected in 14 metagenomes. Encoding a membrane-associated form of this enzyme, with a C-terminal cytochrome *c* domain[42], this gene was found on Burkholderiales-assigned scaffolds in all cases where detected. Genes encoding the remaining steps in the uric acid degradation pathway (i.e., 5-HIU→OHCU→allantoin→allantoate→urea) also classified to Burkholderiales (Supplementary Fig. 7). In total, our analyses suggested complete gene sets for this pathway, within this taxon, in 13 out of 18 metagenomes (Supplementary Data 1; Fig. 4; Supplementary Fig. 8). Maximum likelihood phylogenies of the bacterially encoded PuuD and UraH proteins revealed monophyly of homologs from *Cephalotes*-associated Burkholderiales (bootstrap support = 98% for PuuD and 76% for UraH; Supplementary Fig. 11). The topology of both trees suggested relatedness to homologous proteins from free-living Burkholderiales and other Proteobacteria.

While the genes synthesizing urea from uric acid mapped to numerous scaffolds in several metagenomes, in seven libraries, they mapped to just one or two Burkholderiales-assigned scaffolds. Synteny was conserved in these cases, and such scaffolds also possessed additional genes encoding the subunits of xanthine dehydrogenase (Supplementary Fig. 7), an enzyme converting xanthine into uric acid. Core symbionts, in turn, appear to produce xanthine via purine recycling: genes for guanine deaminase enzymes (Supplementary Fig. 12) were classified to Burkholderiales bins in 16 metagenomes (Supplementary Data 1). Adenine deaminase enzymes were similarly encoded by bacteria, primarily Rhizobiales, across 10 metagenomes.

Further analyses of our metagenomes revealed that various bacteria can produce urea through mechanisms other than uric acid breakdown (Fig. 4; Supplementary Data 1). For instance, across all hosts, microbes from the Burkholderiales, Rhizobiales, Xanthomonadales, and/or Pseudomonadales possessed arginase genes, catalyzing a reaction that converts arginine to urea and ornithine. In several metagenomes, arginase genes also binned to Hymenoptera, suggesting their presence in *Cephalotes* genomes. Genes for a separate, two-step pathway converting arginine to urea (Supplementary Fig. 13) were present in most metagenomes; only Burkholderiales consistently encoded both steps.

The synthesis of uric acid via symbiont purine metabolism further implicates the microbiome in the efficiency of the *Cephalotes* N economy. Such efficiency is further evidenced by our findings of urea synthesis, via uric acid and arginine, by symbionts, and in the latter case, by hosts. Additional metabolic features of host ants may further govern the predominant forms of N waste reaching gut-associated N recyclers. In particular, the

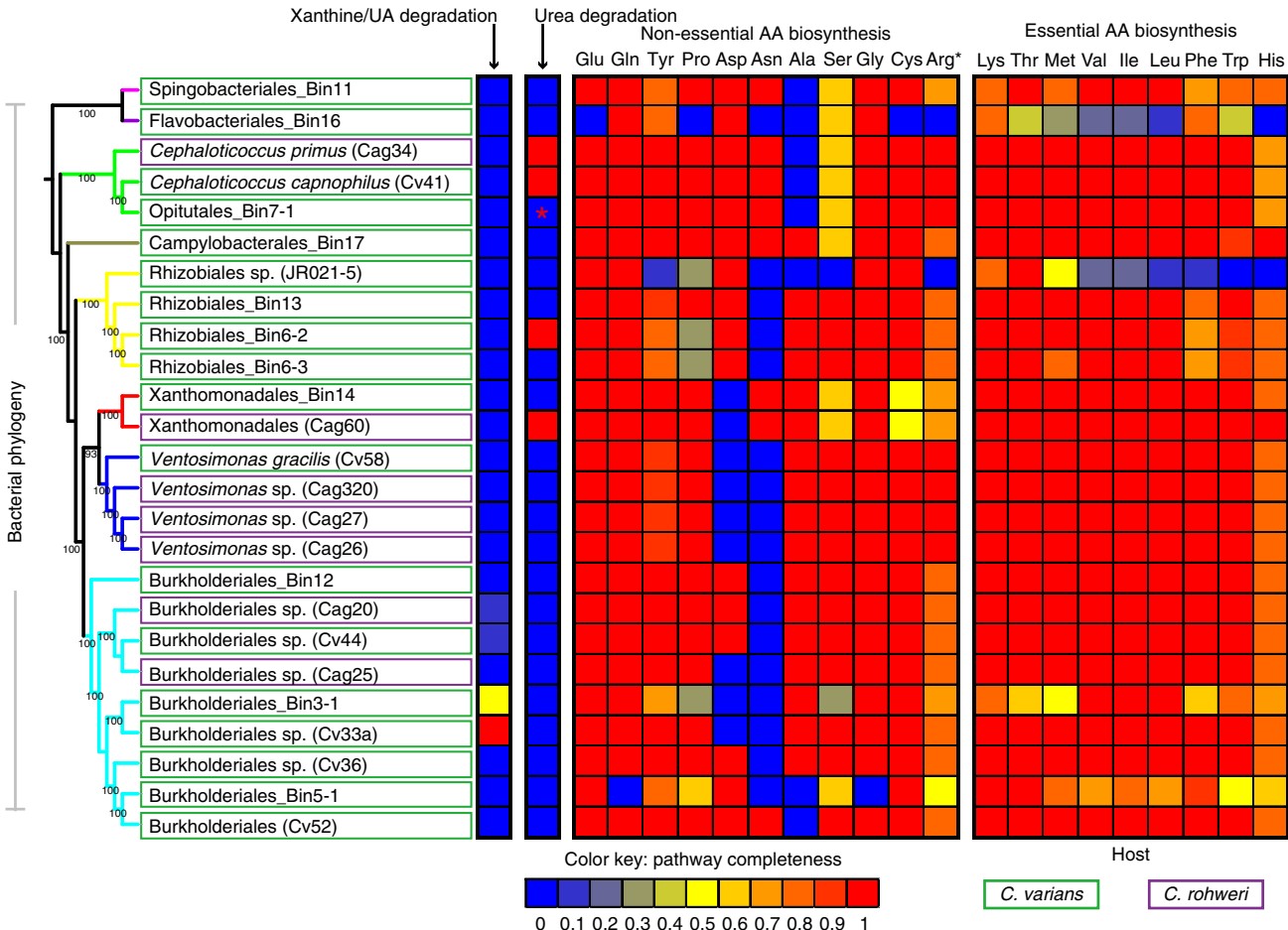

**Fig. 5** Core symbiont strains possess complete or near complete pathways for N-recycling and amino acid biosynthesis. Heatmap illustrates the proportion of genes present from each N-metabolic pathway across distinct symbiont strains. Coding capacities for strains were inferred from 14 fully sequenced cultured isolate genomes (symbionts from *C. varians* and *C. rohweri*) and 11 draft genomes (assembled from *C. varians* colony PL010 metagenome; identified by the term "Bin" within their names). The maximum likelihood phylogeny of symbiotic bacteria on the left was inferred using an alignment of amino acids encoded by seven phylogenetic marker genes obtained from symbiont genomes, and branch colors are used to illustrate distinct bacterial orders. Red asterisk for urea recycling in the *Cephaloticoccus*-like Opitutales bin (7-1) indicates that urease genes from the PL010 metagenome binned to Opitutales, but not to the draft genome for the dominant strain. When combined with the likely presence of just one Opitutales strain within the PL010 microbiome (see Results), it is likely that a completely assembled genome would encode all urease genes. Black asterisk for arginine indicates that we examined the completeness of arginine biosynthesis based on the pathway with glutamate as a precursor

presence of genes encoding uricase (*uaZ*) and 5-hydroxyisourate hydrolase (*uraH*) enzymes on Hymenoptera-assigned scaffolds (Supplementary Data 1; Supplementary Fig. 14) implicates *Cephalotes* ants in the partial breakdown of uric acid. Future research on the relative importance of external N-waste delivery to symbionts (via ant metabolism or diet), the identities of the delivered metabolites, and the roles of symbionts in deriving their own N-wastes for recycling, will be key to a comprehensive understanding of N flow within the *Cephalotes* holobiont.

**Refining symbiont role assignments**. Multiple strains for many of the aforementioned core symbiont taxa often co-exist within a single gut community[37]. So despite pathway "completeness" assessed at the level of host order, it remains unclear whether individual symbionts encode complete pathways for key aspects of N metabolism. We addressed this limitation through genome sequencing for 14 cultured symbionts isolated from two distantly related *Cephalotes* species, *C. varians* and *C. rohweri*. De novo assembly of the 14 shotgun sequencing libraries resulted in assembled genome sizes ranging from 1.9 Mbp to 3.4 Mbp

(Supplementary Table 5). These targeted isolates hailed from widely distributed, cephalotine-specific lineages that spanned five of the eight orders containing core symbiont taxa. They were identical or nearly identical at 16S rRNA gene sequences in comparison to abundant symbiont strains sampled using culture-independent methods (Supplementary Fig. 6). Furthermore, alignments of their genomes to symbiont-derived scaffolds from *C. varians* metagenomes (Supplementary Fig. 15) confirmed their close relatedness to abundant symbionts from the *Cephalotes* gut habitat. These comparisons combine to support the relevance of our below in vitro findings, given the abundance of the studied strains—or of their very close relatives—in vivo.

Highlights of this work (Fig. 5; Supplementary Data 2) included the discovery of a Burkholderiales strain (Cv33a) with a capacity to convert uric acid into urea. While this isolate lacked urease genes, suggesting an inability to convert the derived urea into ammonia, three other cultured symbionts encoded all genes necessary for urease function, including *Cephaloticoccus* isolates from *C. varians* (Cv41) and *C. rohweri* (Cag34), and a Xanthomonadales symbiont from *C. rohweri* (Cag60). Out of

14 isolates, 13 encoded the glutamate dehydrogenase gene (*gdhA*) converting ammonia into glutamate, and most encoded complete pathways for synthesizing most amino acids. As expected from metagenomic analyses, all genomes lacked nitrogenase genes.

Difficulty in cultivating symbionts from three core orders—Campylobacterales, Flavobacteriales, and Sphingobacteriales—limited our ability to infer strain functions for common core

lineages. Insights for these groups were gained through draft genome assembly from our best-sampled metagenome (i.e., *C. varians* colony PL010) using the Anvi'o platform (version 1.2.3)[43] in conjunction with the CONCOCT differential coverage-based binning program[44]. The 11 near complete draft genomes, where >87% of universal single copy genes were detected, spanned seven of the eight orders containing core symbiont lineages, i.e., all but

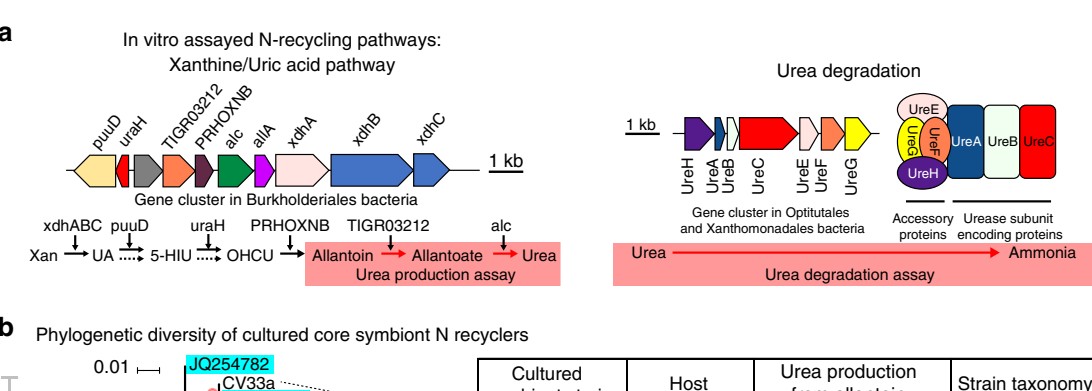

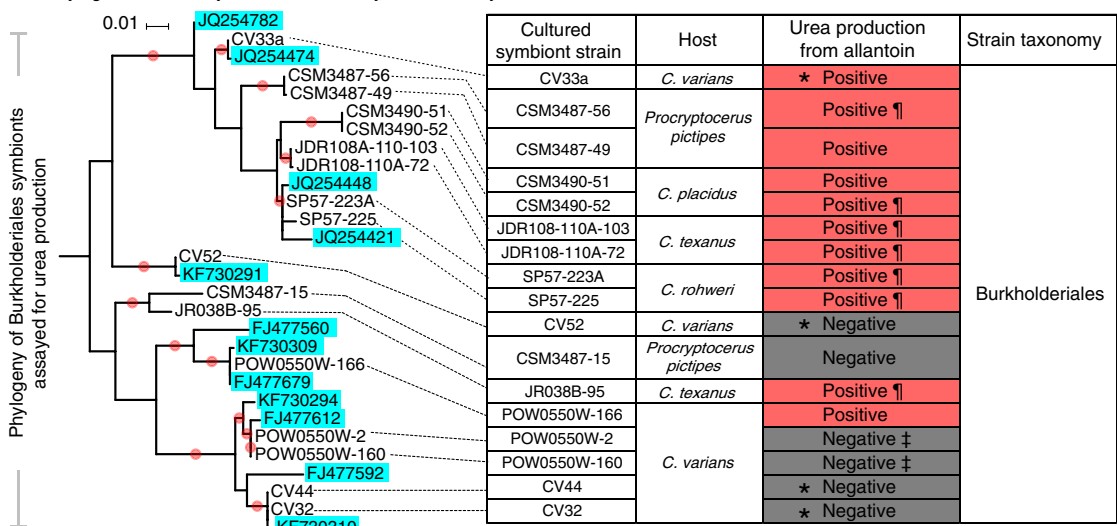

¶ Urea is produced without allantoin but allantoin boosts urea production.
‡ Urea is produced and allantoin has no impact on urea produciton.

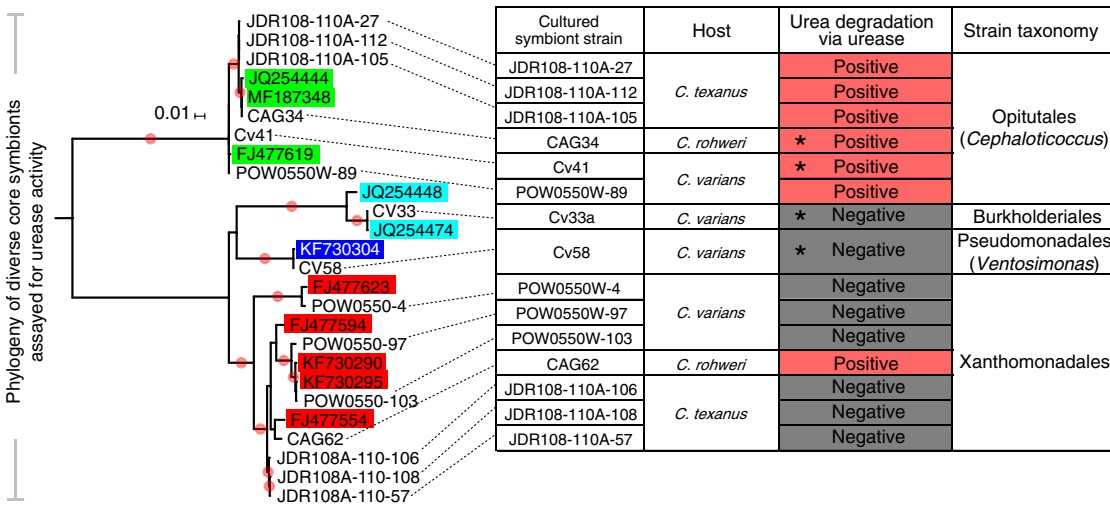

Core cephalotini bacteria from culture-independent efforts

● ≥80% Bootstrap support

Burkholderiales ■ Opitutales ■ Pseudomonadales ■ Xanthomonadales

★ Isolates with sequenced genomes: in all cases in vitro results match expectations from genome annotation.

Pseudomonadales (Fig. 5; Supplementary Tables 6; Supplementary Data 3, 4). Gene content analyses supported findings from metagenomics and cultured isolate genomes. In short, the dominant symbiont strains individually encoded up to 17 complete amino acid biosynthesis pathways. Incomplete pathways were often missing just one or two genes. Nearly all draft genomes showed capacities to assimilate ammonia into glutamate.

N-recycling pathways appeared incomplete within strains from the Burkholderiales and Opitutales (genus *Cephaloticoccus*) showing close relatedness to cultured strains that possessed complete recycling capacities. This could suggest the occasional absence of these functions from these taxa. However, assays on cultured symbionts (Fig. 6) suggest that uric acid or urea recycling capabilities may be ubiquitous within such lineages, raising the possibility that key genes were missing from these draft genomes due to challenges of scaffold binning. Consistent with this explanation was our finding that urease genes were indeed present in the metagenome used to generate these draft genomes (PL010), and that these genes were encoded on abundant scaffolds assigned to the Opitutales (Supplementary Data 1; Supplementary Fig. 7). Furthermore, genes from Opitutales involved in various forms of N metabolism were typically present in just a single copy within this metagenome (Supplementary Data 1). The presence of a single 16S rRNA gene copy from this same taxonomic bin further suggested the presence of just one Opitutales strain in the PL010 colony. The simplest resulting explanation is that Opitutales-assigned urease genes that were not assigned to a draft genome indeed belonged to the Opitutales strain with the assembled draft genome.

**Verifying symbiont roles through in vitro N-recycling assays.** To test whether genetic signatures reflect actual N-recycling capacities, and to study the conservation of this role within key symbionts, we performed a series of in vitro assays from isolates cultured from an expanded number of cephalotine species. Urease activity was qualitatively assessed by the generation of ammonia in the presence of urea, and we obtained positive results for 7 of 15 tested isolates (Fig. 6). All *Cephaloticoccus* (Opitutales) were positive, as was one of six Xanthomonadales isolates. Results for four isolates with sequenced genomes accurately reflected predictions from the presence/absence of urease genes.

Production of urea from allantoin served as a proxy for the activity of the xanthine/uric acid degradation pathway (Fig. 6). Urea was produced from allantoin for 11 of the 17 assayed Burkholderiales isolates (Supplementary Data 5), suggesting function for at least a part of this pathway. Coding capacity

from the five isolates with sequenced genomes accurately predicted the results of these assays.

In summary, genomic inferences on N-recycling seem to accurately reflect the metabolism of core symbionts. And importantly, the phylogenetic placement of strains with in vitro assay data reveal limited distributions of uric acid and urea recycling, with notable enrichment in two clades—*Cephaloticoccus* and a specific, unnamed lineage of Burkholderiales. This suggests long-standing roles for these symbionts in the efficient use of N by the *Cephalotes* holobiont.

**An efficient N economy in *Cephalotes* holobionts.** Our proposed model for symbiont contributions toward the *Cephalotes* N economy is presented at the top of Fig. 4. In short, we predict that waste N is acquired by symbionts through a combination of ant metabolism, diet, and symbiont metabolism. Such N is recycled by a limited range of bacteria, prior to being assimilated into amino acids by most members of the microbiome. The majority of abundant symbionts can make most essential amino acids, and it is posited that N entry from waste products into glutamate serves as a gateway for N transfer to the amino acid pool, given glutamate's role as an $NH_3$ donor in numerous transamination reactions (Supplementary Fig. 1).

Numerous aspects of this hypothesized model await testing through direct experimentation. Assessing this model will further require an explanation as to why N from dietary glutamate could have a small relative impact on host amino acid pools (Supplementary Fig. 5), if glutamate-facilitated transamination is major port of entry for symbiont-recycled N into the amino acid pool. Possible explanations include absorption of glutamate across the midgut wall, and thus depletion of this introduced nutrient, before it can reach symbionts, which are concentrated in the posterior midgut and throughout the ileum[19]. It may also be possible that a large influx of waste N (e.g., dietary urea) can activate symbiont metabolism to a greater degree than an influx of non-essential dietary amino acids, such as glutamate. In addition, the capacities for symbionts to derive glutamate internally (i.e., via ammonia assimilation), as inferred from (meta)genomes, raise questions about the necessity of, and investment in, glutamate import by *Cephalotes*' gut bacteria. Future experiments on the identities of nutrients imported and exported by gut symbionts will be informative. Similar efforts should elucidate whether hosts acquire symbiont-derived N by absorption of symbiont-exported amino acids across the gut wall, or through absorption of N acquired through the digestion of symbiont cells.

**Fig. 6** A limited range of cultured core symbiont strains recycle N in vitro. Results summarize findings from symbionts of five cephalotine ant species, including four from the genus *Cephalotes* and one from its sister genus *Procryptocerus*. **a** Genes and pathways used by specialized gut symbionts to recycle the N-wastes uric acid and urea. The conserved architecture for clusters of symbiont N-recycling genes is illustrated. Red boxes within these pathways represent the metabolic steps assayed for (**b**). **b** Shown at left are phylogenies of cultured symbionts subject to metabolic assays in vitro and their closest relatives in the NCBI database, which are highlighted with taxon-specific, colored boxes. Most cultured isolates had 16S rRNA sequences that were identical or highly related to at least one sequence obtained from a *Cephalotes* ant through culture-independent means. Genomes from such isolates also showed high similarity to abundantly represented scaffolds from our metagenomes (Supplementary Fig. 15). Nodes for cultured symbionts are connected to relevant rows within data tables, where the results of assays for urea production and urea degradation assays are illustrated. Asterisks highlight isolates with a sequenced genome; for each of these, in vitro results matched expectations derived from genome content. Additional symbols used in the urea production table indicate whether allantoin boosted urea production and whether urea production was completely allantoin dependent, and hence likely dependent on uric acid metabolism (e.g., urea can separately be produced through arginine metabolism). Three biological replicates were run as a single trial for 13 symbionts subjected to the urea production assay. For four strains we ran one or two additional trials (Supplementary Data 5). Statistical analyses included a two-way, repeated measures ANOVA to examine an effect of allantoin on urea production, an effect of time, and an interaction between these two factors. Holm–Sidak tests were then used for pairwise comparisons between treatments (e.g., allantoin presence vs. absence) at particular time points, for all trials with a significant effect of treatment or a treatment by time interaction effect

## Discussion

Our data show that the ancient, specialized gut bacterial communities of *Cephalotes* ants recycle nitrogenous waste acquired through the diet and, likely, through ant and symbiont waste metabolism. Importantly, workers acquire large amounts of symbiont-recycled N in the form of essential and non-essential amino acids. While we have empirically confirmed the physiological impact of this process in one species, *C. varians*, our metagenomic analyses show that gene content for N metabolism varies little within certain core lineages found across a broad range of *Cephalotes* species. Our efforts further demonstrate that core symbiont isolates, obtained from several unrelated cephalotine hosts, engage in key N-recycling activities in vitro (Fig. 6). Genomes from such symbionts, cultured from two distant hosts (Fig. 5), closely reflect these in vitro- and metagenome-based inferences. When combined, these findings argue for conserved N-recycling roles for core gut symbionts of *Cephalotes* ants. Such conservation appears to span 46 million years of *Cephalotes* evolution, and possibly longer, given discoveries for similar symbiont function in the sister genus, *Procryptocerus* (Fig. 6). Ancient retention of an efficient N economy supports the hypothesis that this multi-partite gut microbiome plays an adaptive role in the exploitation of an N-poor dietary niche.

Despite the apparently widespread occurrence of N-provisioning symbionts across insects, the magnitude of symbiont contributions to host N budgets has rarely been calculated. Data for wood-feeding termites implicate N-fixing bacteria in providing up to 60% of the N in termite colonies[8]. Measures from the leaf-cutter ant system suggest that N-fixing bacterial symbionts provide 45–61% of the fungus garden's N supply[45], suggesting a large contribution toward the N budgets of these ant farmers. Our estimate from *C. varians* suggests that 5 weeks of feeding on waste N was sufficient to enable symbionts to contribute urea-derived N to 15–36% of the free amino acids in host hemolymph. This is clearly substantial, though not directly comparable to the aforementioned estimates, due to experimental differences.

Reduced survival of antibiotic-treated workers, on diets where urea was the only source of N (Supplementary Fig. 3), further suggest the importance of symbiont contributions to N metabolism in adults, and a similar importance for bacteria in workers was previously argued for *Cephalotes atratus*[46]. In carpenter ants, the nutritional endosymbiont *Blochmannia* impacts worker performance, larval and pupal development, and colony growth; and the detriments of *Blochmannia* removal can be partially alleviated by the presence of essential amino acids in the diet[17]. While the impacts of *Cephalotes* worker microbiomes on overall colony performance have not been measured, the nitrogen in workers' storage proteins is implicated in larval nourishment for several ant species[47]. These results suggest a potential for adult-associated, N-provisioning symbionts to shape performance across multiple stages of *Cephalotes* development.

The relative importance of the discovered N recycling to adult versus larval performance awaits exploration in *Cephalotes*, as do questions of whether the importance of N recycling varies across seasons, habitats, or host phylogeny. Regardless of these answers, our findings of ubiquitous N-recycling capacities for the *Cephalotes* microbiome parallel those for *Blochmannia* in the ant tribe Camponotini[17,48], further supporting the hypothesized importance of nutritional symbionts in canopy-dwelling, herbivorous ants[12]. A trend of "convergent interactions"[49] has thus emerged: canopy foraging for N-poor or N-inaccessible foods has evolved separately in association with unrelated, yet functionally similar symbionts—*Blochmannia* in the Camponotini, and the diverse bacteria described here in the Cephalotini. Future work on other ant herbivores and their conserved symbionts[20,26] will assess the generality of such functional convergence. Also of interest will be studies of symbiont-independent strategies for navigating N-poor diets[6,50].

The conserved nature of symbiont community composition across cephalotines is remarkable compared to patterns for many arthropods[51,52], adding to a trend across eusocial insects. Within the termites, for instance, many core symbionts hail from host-specific lineages, revealing ancient, specialized relationships[3]. Among the corbiculate bees, some relationships with gut symbionts date back to 80 million years[53]. However, even for these hosts, occasional symbiont turnover takes place—in association with dietary shifts, for termites[54], and among major phylogenetic divisions in bees[53].

Evolved behaviors have likely preserved partner fidelity in these groups. Among eusocial bees, symbiont transfer takes place within the hive, through a combination of trophallaxis, coprophagy, or contact with nest materials[55,56]. Termite siblings transmit symbionts through oral–anal trophallaxis[57,58]. A similar mode of passage has been noted for *Cephalotes* and *Procryptocerus* ants and for other ant herbivores as well[18,22–24]. Of likely further importance for cephalotines is a fine-mesh filter, enveloping the proventriculus, which can bar the passage of particles as small as 0.2 μM beyond the crop. This filter develops shortly after young adults solicit trophallactic symbiont transfers[22]. Symbionts acquired during early adulthood will, thus, be sealed off within the midgut, ileum, and rectum, with minimal opportunities for subsequent colonization by additional, ingested microbes. These dual drivers of partner fidelity[59] may collectively explain the preservation of an ancient nutritional mutualism and sustained exploitation of N-poor foods by successful canopy herbivores.

## Methods

**Ant collections and sample sizes.** Details on ant collections and the uses of ants from particular locales are presented in Fig. 1 and Supplementary Table 1. For many of these protocols, additional details can be found in the Supplementary Methods. All in vivo assays targeted workers from three separate colonies, and isotope labeling experiments utilized replication within each colony (see Supplementary Data 6 for sample sizes). In these cases, we aimed for three biological replicates per treatment, per colony. Each biological replicate consisted of pooled hemolymph samples derived from 3 to 10 workers, which had been randomly assigned to treatment groups. Due to moderate worker survival, we occasionally utilized just one or two replicates per treatment for a given colony. Researchers were not blinded with regard to treatment when harvesting hemolymph materials; nor were they blinded when quantifying in vitro assays for cultured symbionts, as described further below.

For whole ant, in vivo experiments, data from all replicates were included in our statistical analyses. For our in vitro assays, we performed a single trial with $n = 3$ biological replicates per cultured symbiont strain, running statistics on each trial. Urea production assays, and statistics, were separately run a second or third time for 4 of 17 tested strains (Supplementary Data 5). For one strain (CSM3490-51), results conflicted across trials; we reported the findings from the first assay in Fig. 6. For the second and third strains, results were consistent between assays (Cv33a, JR038B-95). For the fourth strain (SP57-223A), the results were generally consistent, aside from a small difference at the last time point.

**In vivo colony fragment experiments.** Experiments on live ants were performed on *Cephalotes varians* colony fragments collected from the Florida Keys. Acetylene reduction assays were used to assess the capacity for N fixation. To achieve this, we incubated between 72 and 87 adult workers (and also, in some instances, queens, larvae, and pupae) in air-tight syringe chambers loaded with acetylene within hours of field capture. After the start of incubation, air samples were collected from each syringe at 0, 1, 2, 4, 8, and 16 h. They were then analyzed with a gas chromatography-flame ionization detector to quantify levels of acetylene, and to detect the production of ethylene, which would indicate the likely presence of nitrogenase activity. Prior assays on termites with N-fixing bacteria had detected ethylene production within 4 h[60,61]. Sampling across 16 h was, thus, deemed to provide an adequate opportunity for ethylene detection, if abundant *Cephalotes* symbionts were expressing nitrogenase under the study conditions.

Controlled lab experiments were performed to quantify microbial contributions toward N upgrading of non-essential dietary amino acids ($^{15}$N- or $^{13}$C- glutamate experiments) and, separately, to quantify N recycling, and subsequent upgrading, of dietary N waste ($^{15}$N-urea experiment). In each experiment, we utilized $n = 3$ colonies. Workers from each colony were split into three equally sized groups, and

subjected to three separate treatments. In the first treatment, workers were fed antibiotics to suppress or eliminate their gut bacteria for 3 weeks. After this time, workers transitioned to the trial period where they were continuously fed antibiotics in addition to a diet with labeled glutamate (with $^{15}$N or $^{13}$C) or one with labeled urea ($^{15}$N). Feeding for this trial period lasted 4 to 5 additional weeks. Symbionts were undisturbed in the second and third treatment groups. Diets fed to these latter workers during the 3-week pre-trial period were, hence, free of antibiotics but otherwise identical to those of treatment group one. For the 4–5-week trial period, workers from the second treatment group were fed on antibiotic-free versions of the aforementioned heavy isotope diets; those from the third group were fed on the same diet, with the exception that the glutamate or urea consisted of standard isotope ratios (i.e., biased toward lighter isotopes).

For two of our experiments we recorded worker survival throughout the trial period. We noted that antibiotics increased worker mortality on the urea diet, while having no effect on survival in the examined glutamate experiment (Supplementary Fig. 3). Impacts of antibiotic treatments on bacterial titer were quantified using quantitative PCR (qPCR) assays targeting the bacterial 16S rRNA gene; for a subset of specimens, we examined the impacts of antibiotics on community composition, via amplicon sequencing of 16S rRNA (Supplementary Fig. 2; Supplementary Data 7). Worker hemolymph was harvested at the end of the 4–5-week trial, from 3 to 10 surviving workers per colony, with a goal of three replicates per colony, per treatment. Hemolymph was then pooled within each replicate, used for amino acid derivitization, and subjected to GC–MS to quantify proportions of free amino acids containing the heavy isotopes (see details in Supplementary Data 6).

**Metagenomics**. Adult workers were dissected under a light microscope using fine forceps, with removal of gut tissues from each dissected *Cephalotes* worker. DNA extractions were performed on single guts for 10 workers, from each of two colonies of *C. varians*, or on a pool of guts from 10 workers in one colony, for each of the 16 remaining *Cephalotes* species. Separate extractions from *C. varians* siblings were then pooled within each colony to generate the starting genomic material for metagenomics in this species. The resulting two DNA samples and the 16 samples from other *Cephalotes* species were then used for metagenomic library preparation. After size selection, adapter ligation, amplification, and clustering, samples were sequenced ($2 \times 100$ bp or $2 \times 150$ paired end reads) on an Illumina HiSeq2500 machine. Sequences were trimmed for quality, with removal of adapter sequences after de-multiplexing. Assembly of reads from individual libraries then proceeded using a variety of $k$-values with the IDBA-UD metagenomic assembler. Scaffolds were uploaded to the Integrated Microbial Genomes with Microbiome Samples Expert Review (IMG/M-ER) website[62]. Classification in IMG/M-ER proceeded based on USEARCH similarity against all public reference genomes in IMG and the KEGG database.

Scaffolds were occasionally binned to taxa not predicted to be found in the *Cephalotes* gut microbiome. For example, urease gene-encoding scaffolds classified to the order Rhodocyclales. Prior 16S rRNA-based studies using various universal primer pairs had failed to detect Rhodocyclales symbionts in *Cephalotes* ants. Since these particular scaffolds showed similar depths of coverage to scaffolds from Opitutales, and since the encoded urease genes were identical to those found in cultured Opitutales (i.e., *Cephaloticoccus*), we concluded that this result reflected a classification error. To correct for this, and to query other scaffolds with unexpected classifications, we compared such scaffolds to reference genomes from seven cultured bacterial isolates, hailing from core *Cephalotes*-specific lineages.

After taxonomic classification, IMG/M-ER was used to annotate gene content from our scaffolds and taxonomic bins. Based on these annotations, we focused on N metabolism, using KEGG[63] and Metacyc[64] as guides to manually construct degradation pathways for N-waste products and biosynthetic pathways for amino acids. We examined the completeness of the N-waste degradation pathways based on Fig. 6a and the completeness of the amino acid biosynthetic pathways based on Supplementary Figs. 1 and 8 across 18 metagenomes, 14 isolate genomes, and 11 draft genomes (as described below).

Homologous proteins from N-recycling pathways and 16S rRNA genes were extracted from each dataset and used in phylogenetic analyses with closely related homologs identified, using BLAST searches, in the NCBI database. To further aid in understanding taxonomic composition and to illustrate depths of coverage for the taxa in our libraries, we generated "blob plots" based on read mapping to classified scaffolds using BWA[65] and modified scripts from a prior publication[66]. These graphs showed the depth of coverage for each scaffold in relation to our classifications, along with the %GC content, a taxonomically conserved genomic signature that further aided us in our efforts to visualize the diversity of symbionts within microbiomes (Fig. 3).

**Metagenomic binning to generate draft symbiont genomes**. To improve the assignment of metabolic capabilities to individual symbiont strains we used the Anvi'o metagenome visualization and annotation pipeline (version 1.2.3)[43] in conjunction with the CONCOCT differential coverage-based binning program[44]. In doing so, we binned a subset of assembled scaffolds from the metagenomic datasets of *C. varians* colony PL010—the library with best symbiont coverage—into draft genomes belonging to separate symbiont strains. Reconstruction of N-metabolic pathways was then performed to comprehend the range of metabolic capabilities of individual symbionts.

**Genomics and in vitro assays on cultured bacterial isolates**. Gut tissues from *Cephalotes* and *Procryptocerus* worker ants were dissected and macerated using sterile conditions. Contents were then plated on tryptic soy agar plates, and plates were incubated at 25 °C under an atmosphere of normal air supplemented with 1% carbon dioxide in a $CO_2$-controlled water-jacketed incubator. After colony sub-cloning, pure isolate cultures were maintained under these same conditions on the aforementioned plates or in tryptic soy broth. DNA extracted from these cultures was subsequently used for 16S rRNA PCRs to compare isolates to bacteria previously sampled through culture-independent studies. Isolates from *C. varians* and *C. rohweri* (both previously well studied through culture-independent means) were prioritized for genome sequencing when their 16S rRNA sequences were identical or nearly identical to those of symbionts sampled in prior, in vivo studies. Extracted bacterial DNA was used for library preparation, followed by Illumina or PacBio SMRT sequencing. Assembled genomes were uploaded to IMG/ER for annotation, with N-metabolism pathway reconstruction and extraction of genes/proteins for phylogenetics taking place as described above. Alignments between isolate genomes and assembled metagenomic scaffolds were performed using Icarus[67], as implemented in MetaQuast[68] and visualized in Circos[69]. These assisted in our efforts to infer relatedness between cultured isolates and strains found in abundance in vivo, in the host species *C. varians*.

A subset of cultured isolates was subjected to assays to detect ammonia production from urea. Similarly, several cultured isolates were tested to determine whether allantoin, a derivative of uric acid breakdown, could be used to synthesize urea. Methodological details on these assays are described in the Supplementary Methods. As described above for genome sequencing prioritization, strains prioritized for assays were those deemed highly related to specialized core *Cephalotes* symbionts.

**Fluorescence in situ hybridization (FISH)**. The fixed, dissected gut of an adult worker from *Cephalotes* sp. JGS2370 was hybridized with a set of eubacterial probes labeled with AlexaFluor488 and AlexaFluor555 and imaged, following the protocol in the Supplementary Methods, on a Leica M165FC microscope. Fluorescent microphotographs taken using blue and green excitation filters were then merged with a photograph taken under white light.

**Code availability**. Python scripts used in our taxon-associated %GC plots are publicly available at the following urls: https://github.com/sujaikumar/assemblage; https://github.com/DRL/blobtools.

**Data availability**. Raw data from our GC–MS assays and other summarized data are made available in Supplementary Tables published with this manuscript. Assembled genomes and metagenomes are available in IMG under the following accession numbers: Gp0095985, Gp0095983, Gp0125961, Gp0125962, Gp0125963, Gp0125964, Gp0125967, Gp0125968, Gp0125969, Gp0125970, Gp0126569, Gp0126571, Gp0126580, Gp0126572, Gp0126573, Gp0126574, Gp0126575, Gp0126577, Gp0155340, Gp0154034, Gp0154033, Gp0154032, Gp0154031, Gp0154030, Gp0154021, Gp0127963, Gp0110146, Gp0110145, Gp0110144, Gp0110143, Gp0110137, Gp0110136. Information on particular PCR primers and FISH probes are available in the Supplementary Methods. Raw sequence data are available upon request from the authors.

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

## Acknowledgements
We thank Ryuichi Koga for assistance with FISH and Amit Basu for help with amino acid derivitization. Sue Kilham, Shivanthi Anandan, Mike O'Connor, and Sean O'Donnell provided useful suggestions on statistics and experimental design. Pamela Plantinga provided advice on statistical analyses for in vitro symbiont assays. Andrew Nguyen aided in the development of effective antibiotic treatment protocols, and Nancy Moran provided helpful comments on the manuscript. This study was funded by NSF grants 1050360 and 1442144 to J.A.R., NSF grant 1110515 to J.G.S. and N.E.P., NSF grant 1442316 to C.S.M., and NSF grant 1442156 to J.T.W. Funding was also provided by SNSF grant IZK0Z3_164213 to Y.H. and P.E., and by SNSF grant 31003A_160345 to P.E.

## Author contributions
Y.H., J.G.S., and J.A.R. directed the project. Y.H., P.L., and J.A.N. performed in vivo colony fragment experiments. Y.H. and J.G.S. performed molecular biology experiments. Y.H., J.G.S., P.E., Y.L., and C.L.D. performed the bioinformatic analysis. J.G.S. performed (meta)genomic sequencing. J.S.M. performed GC–MS analyses and D.R.V. performed analyses using a gas chromatography-flame ionization detector. J.T.W. and M.S. performed all in vitro assays on cultured bacterial isolates. Y.H., C.S.M., and D.J.C.K. collected ant specimens used in this study. P.L. performed FISH. J.A.R. and Y.H. wrote the manuscript and J.G.S., J.W., P.L., P.E., C.S.M., N.P. and D.J.C.K. contributed to the revision of the manuscript.

## Additional information

**Competing interests:** The authors declare no competing interests.

