## [Peer Review File · Nature Communications]

Reviewers' comments:

Reviewer #1 - Expertise: Ant-microbe symbiosis

The submitted manuscript demonstrates the detection of nitrogen waste recycling by oft-associated bacteria in several species of turtle ants, and their contribution to available essential amino acids that can be used by their host.

Overall, the authors (and there's a bunch of them!) make a convincing case for this using both genomic data and various molecular and in vitro assays that were informed by the 'omics data. This latter point was quite satisfying to see the authors using the 'omics data to develop and perform functional tests.

This is a significant contribution to our understanding about how herbivorous and omnivorous insects collaborate with their gut microbes to obtain (and retain) sufficient nutrients.

Reviewer #2 - Expertise: Nitrogen metabolic pathways in microbe genomes

The aim of this manuscript is to define the function of microbial symbionts in providing available nitrogen to *Cephalotes* ants. There are few studies examining the role of microbial symbionts in providing nitrogen to animal hosts even though this function is quite vital, and ancient, for animals who consume N-poor diets. The study found that microbial symbionts in the ant gut recycle nitrogen from uric acid or urea back to the amino acid pool for essential amino acid production that serves both the microbial community and the ant host.

The findings are novel and extensively validated in this study. The authors sampled ant populations over a large geographical region and performed cultivation experiments, isotope labeling experiments, metagenomic sequencing, genome analysis of symbionts, and activity assays to demonstrate and validate the diversity of N-recycling and amino acid biosynthesis pathways across commensal populations within a large number of different ant species. The experiments very clearly connected the microbial symbiont population with their function in N-recycling and amino acid biosynthesis. The work also dismissed the hypothesis that microbial symbionts fix N_2 as a source of nitrogen for the ants by revealing a complete lack of N-fixation genes and zero acetylene reduction activity.

The physiological experiments were very well controlled (e.g. inclusion of antibiotic-treated ants to demonstrate specific role of microbes in providing C and N to ant amino acid pools). In addition, the replication across many ant species across a large geographical range is commendable. The only part that was a little confusing was the paragraph on lines 216-228. The main point here is that synthesis of all essential and non-essential amino acids are covered by the entire community (metagenome) yet individual taxa lack bits and pieces of various pathways. Is it important to point out the incomplete pathways for each taxon in the text? It takes a couple of readings through this paragraph to grasp that the missing parts of pathways do not matter because there is ample redundancy in other taxa to ensure a high rate of N-recycling.

The work is clearly presented and convincing as it offers multiple lines of evidence all pointing to a specific function of microbial symbiont communities in recycling N from uric acid and urea into amino acid pools for uptake within the ant gut. The experiments are so extensively performed and validated that this reviewer could not think of additional evidence to gather to make the point. The figures are well crafted and will be quite useful to readers and researchers in diverse fields (e.g. microbiology, genomics, ecology, entomology, etc.). It is a hefty contribution with an extensive data set, yet the

authors distilled the information into a digestible, and highly enjoyable, read.

Reviewer #3 - Expertise: Insect microbiomes & genomics

The manuscript by Hu et al explores the relationship of cephalotine ants with nitrogen recycling bacteria. The authors show that members of the order Oplitales are found consistently in association with multiple species of *Cephalotes* spp and that their genomes consistently code for genes capable of recycling urea, a nitrogenous waste compound that could be produced by ants or their microbiota or that could be encountered by the ants as they feed in diverse food sources, such as excreta. The authors also found other bacterial orders with these capacities, but their distribution among *Cephalotes* spp was not consistent. Conversion of urea into several essential amino acids was demonstrated and exposure to antibiotics reduced urease activity and the conversion of urea into these amino acids, supporting the role of the microbial community in recycling urea and provisioning essential amino acids. Overall, the comparative metagenomics analyses and lab experiments support an evolutionarily conserved relationship between Oplitales and *Cephalotes*.

The manuscript is well-written and the analyses included are very supportive of a conserved relationship between turtle ants and nitrogen recycling bacteria, which will certainly be of broad interest to those studying host-microbe interactions. I can recommend publication in *Nature Communications*; however, I have a few major comments that could be addressed as well as some minor comments that would improve the manuscript.

Major comments:

The authors claim that nitrogen fixation is absent from these ants and that any previous associations of nitrogen fixing bacteria with ants was likely due to coincidence or contamination. However, only three ant colonies were tested with acetylene reduction. Was the incubation time suitable to detect fixed nitrogen? Was nitrogenase activity also absent from the 15 other species selected for metagenomic analysis? Likewise, the lack of ability to detect nitrogenase or related genes in metagenome assemblies is not sufficient to conclude that they are completely absent. They could have been present in low abundances and may have not been assembled. However, I would agree with the authors that nitrogen recycling is likely one of the major contributors to nitrogen recycling in this system. Thus, the authors should revise their conclusions accordingly. See line-by-line comments for more specific suggestions.

The authors claim that urea recycling is a major contributor to N economy in *Cephalotes* spp, but have only tested three colonies from one species. Do the authors have evidence to suggest that recycled nitrogen is also used by any of the 15 other species included in the study?

The next major comment is related to the uricase genes detected in the metagenomics assemblies. The authors mention that several OTUs coded for these genes, suggesting that uric acid could also be recycled; however, they also mention that uricase genes were detected on hymenopteran scaffolds. Are the bacteria in the gut contributing to uricase activities? Or is the insect capable of doing this on its own? An experiment comparing antibiotic treated and untreated insects could be used to see if disrupting the microbial community has any impact on this activity in ants could demonstrate whether the microbial community provides or augments this enzymatic activity. The same comment could also apply to the arginase enzymes described later in the results.

Line-by-line comments:

Line 39: I would avoid using the term mutualism because it was not determined whether or not the microbes benefit from this relationship. I would use the term symbiosis here and throughout.

Line 50: This phrase is confusing as written: co-opting symbiont N metabolism for their own benefit. Suggest revising to "co-opting relationships with symbionts that contribute to the N economy."

Line 66: *Camponotini* should be defined in the previous paragraph.

Line 73: To investigate the potential role of symbiotic bacteria in N acquisition,

Lines 77-778: This sentence seems like it is missing a word. Try....urine, bird feces, and other excreta

with...

Line 78: Replace Given this with Given its nitrogen deficient diet...

Line 88: How were the time periods for the acetylene reduction assay chosen? Were they based on previous studies in ants? Is it possible that a longer incubation period could have resulted in the detection of some nitrogenase activity? For example, it may be possible that nitrogen is fixed in small amounts by the microbiota and that this fixed nitrogen does contribute to the nitrogen economy in this system. Therefore, I would not conclude that these findings argue against nitrogen fixation in this system entirely, but rather that recycling may be a more prominent contributor to N economy relative to fixation. Additionally, an N¹⁵ assay, similar to the ¹⁵N-labeled urea and glutamate assays presented by the authors, would provide further support that fixed nitrogen is not a prominent contributor to nitrogen economy in this system.

Lines 93-102: The approximate concentrations of the amino acids that were included in the holidic diet would be useful for those who want to perform similar studies in other ant species. I think this section could use a little bit more explanation about amino acid synthesis---for example, what essential amino acids can be synthesized from glutamate? Additionally, it might be helpful to mention that the insect itself can use glutamate to synthesize several non-essential amino acids and that is why you may not observe many differences between the antibiotic treated and control insects.

Line 115: with a small effect size.

Line 124: To test this hypothesis....

Lines 135-141: Were any of the other Cephalotes species used for metagenomic analysis tested for acetylene reduction and N¹⁵-enrichment with urea, uric acid, or glutamate as the substrates. Such information would help demonstrate the prevalence of the N from recycled urea into amino acids across this genus and would greatly aid in the interpretation of the metagenomic analysis. It would also add further evidence to the author's claim that nitrogen fixation is not a major component of nitrogen economy in this genus.

Line 139: Two of these samples

Line 144: What do you mean here by host-specific clades? Do you mean that the taxonomy of the communities was conserved across Cephalotes, but that each of the Cephalotes harbored different OTUs/genera/etc? I think a few additional details about the results of this previous study would be useful and help with the interpretation of the results from the metagenomic sequencing.

Line 145: Blast analysis of the 200 bp fragments is not the best method for taxonomically classifying 16S sequences. For instance, taxonomic informativeness varies greatly across 16S rRNA. Sequence data from some regions may only bin to higher taxonomic levels while other regions carry more family and genus specific sequence signatures. What measures were taken to control for this variation? What databases were searched? NT? Other? If NT, why wasn't a dedicated 16S database used for taxonomic classification (RDP)? Such databases are often more accurately curated than the sequences housed in NCBI and can carry more reliable taxonomic information. Lastly, what was the % similarity to these previously identified taxa? Figure 3. I am having a hard time finding the N-recycling bacteria in the tree. Consider increasing the weight on those branches to make them stand out more.

Line 155: At what taxonomic rank the microbiome conserved? Class level? Or at finer taxonomic levels (genus, etc)? The authors should clarify those findings in the text.

Line 156: The authors mention that Oplitales was the most dominant order detected in all of the samples. However, in Figure S1 (16S amplicon analysis), that does not always appear to be the case in the three colonies presented in the amplicon analysis. This should be addressed.

Line 159: co-existence of distinct strains....the 10% difference in GC content suggests that the different Xanthomonadales strains that the authors refer to may belong to different genera or families. Suggest using the term 'operational taxonomic unit' (OTU) instead of strain here for that reason. Same comment for line 162 and also throughout the manuscript. Also, it is difficult to read the axes labels on figure 3, so it is hard to determine how different the depths of coverage are for the read clouds. Small differences in depth may not indicate different strains and, likewise, genomic regions can vary in terms of GC content. Being able to more clearly see the level of separation would greatly

aid in the assessing the author's claim that these different read clouds indicate different OTUs. Do the rRNAs pulled from the assembly also support multiple OTUs?

Line 161: though still ubiquitous across metagenomes,

Line 163: I think you can change 'each taxon' to 'each order' for more clarity.

Line 165: It is not clear how presence/absence calls for N metabolism genes suggested rarity of these symbionts? A better indicator would be abundance of 16s rRNA or another universally conserved gene.

Lines 173-176: Absence of nitrogenase genes in the metagenomic assemblies from insects that were not tested for acetylene reduction could simply be due to low abundance. Thus, the authors should not completely exclude the presence of nitrogen fixing bacteria unless more specific PCR-based analysis targeting nifH were also employed (and were also negative).

Line

Line 192: change additional bacteria encoding complete sets to additional complete sets

Lines 213-214: I think this sentence is missing a word or two. Please check. Also, typo....extensive.

Additionally, it is still not clear what the authors mean by 'core taxa.' Do you mean core orders (as in genes involved in these processes were assigned to bacterial orders found in association with all Cephalotes)? Please make sure this term is very clearly defined early on in the results section.

Line 222: It is unclear what the authors are referring to here....'This paralleled.' What does 'this' refer to? Please revise.

Line 235: The authors mention that several genes related to uricase activity as well as one canonical uricase gene were binned to Hymenoptera. Is it possible that the insect can perform these functions on its own without the help of bacteria? It would be interesting to perform a uricase assay on antibiotic treated and controls to see if uricase function declines in treated individuals. I have also noticed that large numbers of insect genomes (Hymenoptera and Coleoptera) contain genes related to uricase function and have wondered if these paths are functional.

Line 256: as genes for guanine deaminase enzymes were identified...

Line 266: metagenomes, but only....

Line 270: Again, I would choose another word besides strains as that implies more than one strain from an individual bacterial species. I would use OTUs instead.

Line 273: We addressed this shortcoming

Lines 273-275: It is not completely clear how the bacterial isolates were selected for full genome sequencing. More explanation is needed here. For example, five of the eight dominant 'taxa'....what does that mean? Dominant orders? Species? Etc? Were the 14 strains selected from these 'taxa'? The authors mention that these core bacteria were shared among two of the ant species analyzed? Were there any core/conserved bacteria/OTUs shared among all of the ant species? Or alternatively, were there few common OTUs? Either observation would be interesting to point out.

Line 277: Were the 16s rRNA genes from the draft genomes also similar to 16s genes in the metagenome assembly? That would lend further proof that the strains selected for sequencing are indeed present in vivo.

Line 283: This sentence is confusing. The strain lacked urease genes but three cultured symbionts encoded all genes necessary for urease function....do you mean three other cultured symbionts coded for urease and all of the necessary accessory genes?

Line 295: What symbiont core taxa are you referring to here?

Line 297: How were the dominant strains identified? By coverage?

Line 307: expanding the number of targeted cephalotine species...suggesting revising this sentence to, expanding the number of targeted cephalotine-associated bacterial species. The way it is worded now makes it sound like you preformed urease assays on the ants themselves (similar to the N15-urea experiments).

Line 328: I would remove the phrase large amounts because the boosts in essential amino acid levels observed by the authors were small under the experimental conditions. It is possible that other nutritional sources could provide amino acids, but I agree that the bacteria are likely an important

component of N acquisition and retention in this system.

Line 355: has thus emerged...I don't think you need the commas surrounding thus.

Line 356: has evolved separately in association with unrelated, yet functionally similar symbionts. Separately within what context? Herbivorous ants in general or within the cephalotine ants? An additional detail or two will clarify.

Line 434: How do you know these scaffold assignments are incorrect? Could these microbes have been consumed with food and be present in the gut? More details would be useful.

794: It is difficult to see the circle sizes in Figure 3 (taxonomy) to indicate the scaffold sizes for each lineage. Also, are these N50 scaffold sizes or total scaffold lengths?

Lines 818-820: It is unclear how the authors are suggesting that there is likely only one Oplitales strain. Please provide more information about why this is likely.

Figure 6: The box colors for the corresponding taxonomic assignments should be defined either on the figure or in the legend.

Reviewer #4 - Expertise: Isotope ecology & labeling experiments

This study tested the hypothesis that Cephalotes ants exploit organic nitrogen derived from their symbiotic microbes. I have a few comments on the manuscript as follows.

L112: Why was the effect size of labelled glutamate small? As illustrated in Fig. 4, Glu is the anchor compound bridging between ammonia and amino acids. I would expect the same results with the urea-feeding experiment. However, Fig. S4 indicates that all amino acids with the exception of Phe were not derived from ^{15}N -labelled Glu. This may suggest that the illustration in Fig. 4 is too much simplified

L128: Why did only asparagine in the heavy urea treatment show insignificant ^{15}N percentage with the controlled treatment, although it is primarily derived from aspartic acid that showed a significant difference? A brief but explicit explanation may be needed here

Fig. 1B: In y axis, delta ^{15}N is not necessarily analogous with trophic level. In theory, it should be calculated from an offset against a given baseline. See Post (2002) Ecology 83:703-718 for more details. Furthermore, Cephalotes ants are no longer "herbivore" and their trophic level should be one unit higher than that of symbiotic bacteria. The authors may be interested in Steffan et al. (2015) PNAS 112:15119-15124

Reviewers' comments:

Reviewer #1 - Expertise: Ant-microbe symbiosis

The submitted manuscript demonstrates the detection of nitrogen waste recycling by oft-associated bacteria in several species of turtle ants, and their contribution to available essential amino acids that can be used by their host.

Overall, the authors (and there's a bunch of them!) make a convincing case for this using both genomic data and various molecular and in vitro assays that were informed by the 'omics data. This latter point was quite satisfying to see the authors using the 'omics data to develop and perform functional tests.

This is a significant contribution to our understanding about how herbivorous and omnivorous insects collaborate with their gut microbes to obtain (and retain) sufficient nutrients.

>>We greatly appreciate the reviewer's positive feedback. The manuscript has required years of work, many revisions, and a very large team of collaborators. So we too are hopeful that all of this effort represents a significant contribution.

Reviewer #2 - Expertise: Nitrogen metabolic pathways in microbe genomes

The aim of this manuscript is to define the function of microbial symbionts in providing available nitrogen to Cephalotes ants. There are few studies examining the role of microbial symbionts in providing nitrogen to animal hosts even though this function is quite vital, and ancient, for animals who consume N-poor diets. The study found that microbial symbionts in the ant gut recycle nitrogen from uric acid or urea back to the amino acid pool for essential amino acid production that serves both the microbial community and the ant host.

The findings are novel and extensively validated in this study. The authors sampled ant populations over a large geographical region and performed cultivation experiments, isotope labeling experiments, metagenomic sequencing, genome analysis of symbionts, and activity assays to demonstrate and validate the diversity of N-recycling and amino acid biosynthesis pathways across commensal populations within a large number of different ant species. The experiments very clearly connected the microbial symbiont population with their function in N-recycling and amino acid biosynthesis. The work also dismissed the hypothesis that microbial symbionts fix N₂ as a source of nitrogen for the ants by revealing a complete lack of N-fixation genes and zero acetylene reduction activity.

The physiological experiments were very well controlled (e.g. inclusion of antibiotic-treated ants to demonstrate specific role of microbes in providing C and N to ant amino acid pools). In addition, the replication across many ant species across a large geographical range is commendable. The only part that was a little confusing was the paragraph on lines 216-228. The main point here is that synthesis of all essential and non-essential amino acids are covered by the

entire community (metagenome) yet individual taxa lack bits and pieces of various pathways. Is it important to point out the incomplete pathways for each taxon in the text? It takes a couple of readings through this paragraph to grasp that the missing parts of pathways do not matter because there is ample redundancy in other taxa to ensure a high rate of N-recycling.

The work is clearly presented and convincing as it offers multiple lines of evidence all pointing to a specific function of microbial symbiont communities in recycling N from uric acid and urea into amino acid pools for uptake within the ant gut. The experiments are so extensively performed and validated that this reviewer could not think of additional evidence to gather to make the point. The figures are well crafted and will be quite useful to readers and researchers in diverse fields (e.g. microbiology, genomics, ecology, entomology, etc.). It is a hefty contribution with an extensive data set, yet the authors distilled the information into a digestible, and highly enjoyable, read.

>>We thank the reviewer for their time, and certainly appreciate their positive assessment of our work. We agree that the section in lines 216-228 could have been more clear. We have re-worked the passage from lines 212-229 to address this concern, as follows:

“Metagenome-derived predictions of functional capacity met these expectations in *C. varians* and all 16 other host species. But in contrast to our findings of one to three urea recyclers per host, genes involved in ammonia assimilation and amino acid biosynthesis were assigned to all core symbiont taxa, suggesting extensive metabolic redundancy. Within all metagenomes, numerous taxa encoded complete gene sets for ammonia assimilation (e.g. Figs. 4, S7, S9, S10; Table S5). Similarly, complete gene sets for the synthesis of each essential and non-essential amino acid were found in bacteria from all metagenomes, though a small number of genes were missing from certain taxonomic bins (Figs. 4, S6, S7, S9, S10; Table S5). Some of these missing genes may have stemmed from low sequencing coverage; as in *C. angustus*. The metagenome from this host had the lowest depths of microbial sequencing coverage in our dataset (Fig. 2), and also contained Xanthomonadales and Burkholderiales bins with an abnormally large number of “missing” genes in several biosynthetic pathways (Figs. 4, S7). The absence of genes could also stem from limits to current pathway annotations, as non-homologous enzymes can converge upon the same functions⁴⁶. It could further arise due to metabolite exchange among hosts or microbes^{47, 48}. Either of these latter two explanations could explain why a single biosynthetic step seemed consistently absent for methionine biosynthesis in Rhizobiales and Pseudomonadales bins, or why the nearly-complete histidine biosynthesis pathways recovered in Xanthomonadales and Burkholderiales bins were consistently “missing” the same step (Figs. 4; S7).”

The manuscript by Hu et al explores the relationship of cephalotine ants with nitrogen recycling bacteria. The authors show that members of the order Opitutaes are found consistently in association with multiple species of Cephalotes spp and that their genomes consistently code for genes capable of recycling urea, a nitrogenous waste compound that could be produced by ants or their microbiota or that could be encountered by the ants as they feed in diverse food sources, such as excreta. The authors also found other bacterial orders with these capacities, but their distribution among Cephalotes spp was not consistent. Conversion of urea into several essential amino acids was demonstrated and exposure to antibiotics reduced urease activity and the conversion of urea into these amino acids, supporting the role of the microbial community in recycling urea and provisioning essential amino acids. Overall, the comparative metagenomics analyses and lab experiments support an evolutionarily conserved relationship between Opitutaes and Cephalotes.

The manuscript is well-written and the analyses included are very supportive of a conserved relationship between turtle ants and nitrogen recycling bacteria, which will certainly be of broad interest to those studying host-microbe interactions. I can recommend publication in Nature Communications; however, I have a few major comments that could be addressed as well as some minor comments that would improve the manuscript.

>>We very much appreciate the reviewer's consideration, and will do our best to respond to each of these comments.

Major comments:

The authors claim that nitrogen fixation is absent from these ants and that any previous associations of nitrogen fixing bacteria with ants was likely due to coincidence or contamination. However, only three ant colonies were tested with acetylene reduction. Was the incubation time suitable to detect fixed nitrogen? Was nitrogenase activity also absent from the 15 other species selected for metagenomic analysis? Likewise, the lack of ability to detect nitrogenase or related genes in metagenome assemblies is not sufficient to conclude that they are completely absent. They could have been present in low abundances and may have not been assembled. However, I would agree with the authors that nitrogen recycling is likely one of the major contributors to nitrogen recycling in this system. Thus, the authors should revise their conclusions accordingly. See line-by-line comments for more specific suggestions.

>>We appreciate the challenge inherent in providing evidence of absence. However, given that nitrogen fixation has to our knowledge never been demonstrated in ants, we rather think the idea that nitrogen fixation ever plays an important physiological role in ants should remain an alternate rather than a null hypothesis. We provide more details below, in responses to the Reviewer's specific comments related to nitrogen fixation. But to first address their direct questions from above:

-We believe the incubation time was sufficient, as we had chosen incubation times consistent with those used in prior studies.

-Yes, nitrogenase genes appeared absent from all 17 *Cephalotes* species based on IMG annotations across all metagenomes (Table S5). In addition, we also performed tBLASTx and BLASTn searches against the most deeply sampled metagenome here, that for *Cephalotes varians* colony PL010. As queries we used previously published *nifH* genes identified from a previous study (Russell et al. 2009). No significant hits were obtained. We have added this information to the results section, immediately after we describe the lack of nitrogenase genes based on IMG annotation.

The authors claim that urea recycling is a major contributor to N economy in *Cephalotes* spp, but have only tested three colonies from one species. Do the authors have evidence to suggest that recycled nitrogen is also used by any of the 15 other species included in the study?

>>Certainly, additional functional confirmation of nitrogen recycling in additional *Cephalotes* species is worth doing. We would argue, however, that it is beyond the scope of what we can accomplish for this manuscript. The primary evidence we have for the ubiquity of this functionality is: 1) the broad conservation of functional genomic potential across the 16 other *Cephalotes* species we have surveyed; 2) in vitro assays show that urease activity, and the ability to make urea from uric acid derivatives, are conserved functions in strains of dominant, core symbionts obtained from unrelated *Cephalotes* species (and from the sister genus, *Procryptocerus*). We have made a note to further emphasize this in the first paragraph of the discussion:

“While we have empirically confirmed the physiological impact of this process in one species, *C. varians*, our metagenomic analyses show that gene content for N-metabolism varies little within certain core lineages found across a broad range of *Cephalotes* species. Our efforts further demonstrate that core symbiont isolates, obtained from several unrelated cephalotine hosts, engage in key N-recycling activities in vitro (Fig. 6). Genomes from such symbionts, cultured from two distant hosts (Fig. 5), closely reflect these in vitro- and metagenome-based inferences. When combined, these findings argue for conserved N-recycling roles for core gut symbionts of *Cephalotes* ants. Such conservation appears to span 46 million years of *Cephalotes* evolution, and possibly longer, given discoveries for similar symbiont function in the sister genus, *Procryptocerus* (Fig. 6). Ancient retention of an efficient N-economy supports the hypothesis that this multi-partite gut microbiome plays an adaptive role in the exploitation of an N-poor dietary niche.”

In the Discussion we also raise a caveat to address the reviewer’s concern: “The relative importance of the discovered N-recycling to adult versus larval performance awaits exploration in *Cephalotes*, as do questions of whether the importance of N-recycling varies across seasons, habitats, or host phylogeny. Regardless of these answers, our findings of ubiquitous N-recycling capacities for the *Cephalotes* microbiome parallel those for *Blochmannia* in the ant tribe Camponotini,^{19, 53} further supporting the hypothesized importance of nutritional symbionts in canopy-dwelling, herbivorous ants¹⁴.”

We hope that these new and modified passages make it clear which lines of evidence support our arguments, and that they have adequately stated where more work will be helpful.

At the same time we would also point out that 10 years after the first experimental demonstration of amino acid provisioning (via N-recycling) by Blochmannia symbionts of Camponotus floridanus, there have been no follow-up studies to further examine this function in vivo that we are aware of. Genomes for Blochmannia symbionts have been obtained from a more limited set of camponotine hosts than those sampled across Cephalotes in this study (i.e. only 6 host camponotine species with sequenced Blochmannia, reported in three separate publications; versus 17 species with sampled metagenomes in Cephalotes—all from our study). In spite of these limitations, the symbiosis and microbiome fields generally “accept” that Blochmannia play a role in N-provisioning across this group.

More broadly speaking, few studies focusing on insect-microbiota interactions have carried out isotope labelling experiments to probe the contributions of the microbiota to host metabolism, helping to make our efforts somewhat exceptional. Given that our study presents one of the most complete shotgun metagenomic analyses for any group of insects (e.g. even well-studied symbionts of corbiculate bees have metagenomes available for only Apis mellifera), the multiple lines of evidence for conserved functional traits in Cephalotes’ microbiota suggest their relevance for the lifestyle of these hosts.

The next major comment is related to the uricase genes detected in the metagenomics assemblies. The authors mention that several OTUs coded for these genes, suggesting that uric acid could also be recycled; however, they also mention that uricase genes were detected on hymenopteran scaffolds. Are the bacteria in the gut contributing to uricase activities? Or is the insect capable of doing this on its own? An experiment comparing antibiotic treated and untreated insects could be used to see if disrupting the microbial community has any impact on this activity in ants could demonstrate whether the microbial community provides or augments this enzymatic activity. The same comment could also apply to the arginase enzymes described later in the results.

>>We agree that this is an interesting puzzle, and certainly not one we have yet completely solved. We do note that we have verified urea production from allantoin in a number of isolates cultured from Cephalotes guts (Table S11). Future experiments to better characterize the fate of uric acid in vivo are planned, but regrettably not feasible for incorporation into this manuscript.

In response to this comment, we have modified the text to better emphasize these unknowns, spelling out a need for more experimental work:

“The synthesis of uric acid via symbiont purine metabolism further implicates the microbiome in the efficiency of the Cephalotes N-economy. Such efficiency is additionally evidenced by our findings of urea synthesis, via uric acid and arginine, by both hosts and symbionts. Additional metabolic features of host ants may further govern the predominant forms of N-waste reaching gut-associated N-recyclers. In particular, the presence of genes encoding uricase (uaZ) and 5-hydroxyisourate hydrolase (uraH) enzymes on Hymenoptera-assigned scaffolds (Table S5; Fig. S11) implicates

Cephalotes ants in the partial breakdown of uric acid. Future research on the relative importance of external N-waste delivery to symbionts (via ant metabolism or diet), the identities of the delivered metabolites, and the roles of symbionts in deriving their own N-wastes for recycling will be key to a more detailed understanding of N flow within the Cephalotes holobiont."

Line-by-line comments:

Line 39: I would avoid using the term mutualism because it was not determined whether or not the microbes benefit from this relationship. I would use the term symbiosis here and throughout.

>>We understand the reviewers' perspective, but we respectfully disagree with their proposed limits to the use of this term. Nutritive endosymbioses of insects (with which this relationship bears many similarities) are typically considered mutualisms; and relying on an empirical determination of fitness effects or fitness proxies for ancient, tightly associated host-microbe interactions poses a critical problem: i.e. benefit with respect to what alternative? If these bacteria are exclusively confined to Cephalotes guts (data suggest, so far, that they are); and if no Cephalotes species lack these bacteria (again, fitting with present data), then there are no alternatives, making it likely that neither party (host or microbiome) can exist without the other.

While we are comfortable with our wording, and would prefer to keep it, we also note that we have not leaned heavily on this terminology in the text. Indeed, we have used "mutualism" only in a general sense once at the end of the abstract and once at the end of the discussion. And we do so to frame the studied host/microbe relationships explicitly in terms of their function, i.e. a 'nutritional mutualism'.

Line 50: This phrase is confusing as written: co-opting symbiont N metabolism for their own benefit. Suggest revising to "co-opting relationships with symbionts that contribute to the N economy."

>>The suggested revision departs from our intention, so we would prefer to leave as-is. We are using co-opt in the sense of 'Divert to or use in a role different from the usual or original one' to refer to herbivore taxa using symbiont N-metabolism in a role different from its original one (i.e. supplying N exclusively to the microbe).

Line 66: Camponotini should be defined in the previous paragraph.

>>We have updated as suggested, introducing this tribe designation immediately after introducing carpenter ants.

Line 73: To investigate the potential role of symbiotic bacteria in N acquisition,

>>We thank the reviewer for this suggestion. But we would prefer to leave as-is since the suggested version seems redundant to the previous sentence.

Lines 77-778: This sentence seems like it is missing a word. Try....urine, bird feces, and other excreta with....

>>We appreciate the comment. This has been now been clarified: “Cephalotes also consume mammalian urine and bird feces, which both contain large quantities of waste N accessible only through the aid of microbes.”

Line 78: Replace Given this with Given its nitrogen deficient diet....

>>Thanks to the reviewer for pointing out the room for improvement here. To reduce redundancy, we have restructured this sentence as, “The remarkably conserved gut microbiomes of cephalotines have been proposed as an adaptation for these N-poor and N-inaccessible diets.”

Line 88: How were the time periods for the acetylene reduction assay chosen? Were they based on previous studies in ants? Is it possible that a longer incubation period could have resulted in the detection of some nitrogenase activity? For example, it may be possible that nitrogen is fixed in small amounts by the microbiota and that this fixed nitrogen does contribute to the nitrogen economy in this system. Therefore, I would not conclude that these findings argue against nitrogen fixation in this system entirely, but rather that recycling may be a more prominent contributor to N economy relative to fixation. Additionally, an N¹⁵ assay, similar to the ¹⁵N-labeled urea and glutamate assays presented by the authors, would provide further support that fixed nitrogen is not a prominent contributor to nitrogen economy in this system.

This is a good question; and the suggestion that we can't completely rule out N-fixation is reasonable. To address the question about our experimental design, we chose the time periods based on previous studies on termites. There are two publications (Pandey et al (1992) Virginia Journal of Science 12, 333–338; French et al. (1976) J Gen Microbiol 96: 202–206) showing no detectable ethylene within the first hour of test for some species, but ethylene can be detected after 4-hour incubation for all surveyed termite species. Thus, we have extended the incubation time to 16 hours to have enough time to detect ethylene.

At the end of this paragraph, we have clarified the text to emphasize the limitations of interpretation inherent in the absence of evidence for a particular metabolism; we also point out the duration of the incubation here, and have now also alluded to our rationale for the chosen time duration in the methods. Here is the section from the Results: “In three separate experiments no detectable ethylene was produced after 16 hours of ant incubation in the presence of acetylene (Table S2). This finding resembled that from a prior experiment on lab-reared *C. varians*, which utilized smaller numbers of workers¹⁷. Collectively, both argue against active and substantial N-fixation by *Cephalotes*' symbionts.”

We have also modified the Methods section to provide more detail on the assay: “To achieve this, we incubated between 72 and 87 adult workers (and also, in some instances, queens, larvae, and pupae) in air-tight syringe chambers loaded with acetylene within hours of field capture. After the start of incubation, samples were collected from each syringe at 0, 1, 2, 4, 8, and 16 hours. They were then analyzed with a gas chromatography–flame ionization detector to quantify levels of acetylene, and to detect the production of ethylene, which would indicate the likely presence of nitrogenase activity. Prior assays on termites with N-fixing bacteria had detected ethylene production within four hours⁶⁹,⁷⁰. Sampling across 16 hours was, thus, deemed to provide an adequate opportunity for ethylene detection, if abundant *Cephalotes* symbionts were, indeed, expressing nitrogenase under the study conditions.”

As noted above, the case for N-fixation in *Cephalotes* has rested entirely on high-cycle PCR detection, and never any actual functional assays. Given this, and given the many different species whose metagenomes failed to yield evidence for nitrogenase genes, we would argue that N-fixation is not a major function of *Cephalotes* gut bacteria. Notwithstanding, we hope the aforementioned changes and the one listed below adequately communicate that we have not completely ruled out the presence of nitrogenase: i.e. the Results subsection titled “Metagenomic analyses: urease is ubiquitous, N₂ fixation is absent” now reads “Metagenomic analyses: urease is ubiquitous, N₂ fixation appears absent”.

Lines 93–102: The approximate concentrations of the amino acids that were included in the holidic diet would be useful for those who want to perform similar studies in other ant species. I think this section could use a little bit more explanation about amino acid synthesis--for example, what essential amino acids can be synthesized from glutamate? Additionally, it might be

helpful to mention that the insect itself can use glutamate to synthesize several non-essential amino acids and that is why you may not observe many differences between the antibiotic treated and control insects.

>>To answer the first question, we have added the following to this section. “Ingredients in the artificial diet consisted of vitamins, minerals, salts, growth factors, carbohydrates, and amino acids. The original recipe⁴² was modified so that only non-essential amino acids were included (Tyr, Asp, Asn, Ser, Ala, Cys, Gly, Glu, Gln, and Pro, but not Arg), each at a 200 mg/ml concentration.”

The second suggestion makes sense, and we agree that there is a benefit to discussing the fate of labeled C and N from glutamate. The possibilities, however, are complex. Arginine and proline are two non-essential amino acids directly synthesized from the carbon skeletons of glutamate. Indeed, these two amino acids showed enrichment for ¹³C when ants were fed diets with heavy glutamate, whether symbionts were there or not. Beyond this, glutamate can be transformed into various compounds through more convoluted metabolism (e.g. via glutamate conversion into alpha-ketoglutarate, followed by entry into the TCA cycle, and subsequent use of TCA products to make Asp, Asn, Met, Thr, Lys, Ile). Note that we show the direct synthesis/carbon skeleton transformations in Fig. S9 (i.e. Arg & Pro synthesis), but not the more convoluted paths for glutamate C transfer (e.g. diversion into the TCA cycle and subsequent transformations).

Transamination using NH₃ from glutamate can result in numerous amino acids acquiring N from glutamate. Such reactions are indicated with triangles falling in the middle of connecting pathway lines in Fig. S9. There are numerous such reactions, and animals can often encode the enzymatic machinery required for such amino group transfers.

Due to this complexity, we have invoked some, but not all, of the specifics in our modified passage. We hope that this strikes a good balance: “Heavy isotope enrichment in the free amino acid pools from worker hemolymph was assessed via GC-MS (Table S3). This allowed us to quantify the use of C or N from glutamate to synthesize host-associated amino acids. Diverse metabolic pathways enable the use of glutamate-derived C or N. Some are encoded by animals and bacteria alike, such as those for non-essential amino acid biosynthesis, and transamination reactions, where the NH₃ group from glutamate is donated to the carbon skeletons of amino acid precursors (Fig. S9). In spite of this complexity, symbiont contributions to host amino acid pools, via dietary glutamate, would be partially evidenced if elevated ¹³C or ¹⁵N signal within ant hemolymph (i.e. on heavy glutamate diets) were eliminated through antibiotic treatment. Results from the ¹³C experiment would be most telling, in many ways. For example, after diversion through the TCA cycle, glutamate-derived carbon could be used by bacteria to synthesize isoleucine, lysine, methionine, and threonine. By definition, these essential amino acids are nutrients that animals cannot synthesize de novo. Hence, any elevation in their ¹³C signal on the heavy glutamate diet, coupled with a lack of such enrichment under antibiotic treatment, would directly implicate symbionts in contributing to host N budgets through the use of dietary glutamate.”

At the end of this section we have also added the following passage, helping to emphasize which lines of evidence most directly support symbiont contributions, albeit small ones: “To summarize, findings that three host-associated essential amino acids (isoleucine, leucine, and threonine) were impacted by

symbiont presence suggests some role for the gut microbiome in upgrading non-essential amino acids from the diet. But small effect sizes raise questions on the importance of such activities to overall host N-budgets.”

Line 115: with a small effect size.

>>We have changed as suggested.

Line 124: To test this hypothesis....

We have changed as suggested.

Lines 135-141: Were any of the other *Cephalotes* species used for metagenomic analysis tested for acetylene reduction and N15-enrichment with urea, uric acid, or glutamate as the substrates. Such information would help demonstrate the prevalence of the N from recycled urea into amino acids across this genus and would greatly aid in the interpretation of the metagenomic analysis. It would also add further evidence to the author’s claim that nitrogen fixation is not a major component of nitrogen economy in this genus.

>>We agree that functional assays from additional *Cephalotes* species will be of value, and plan to perform such assays as opportunity permits in the future. However, performing additional labor-intensive experiments with live insects is outside the scope of what we can do for this manuscript. As explained before, we modified the manuscript wording to emphasize that while our experiments directly demonstrate symbiont function in only one species, the following lines of evidence add up to strongly suggest a conserved N-recycling function across the genus and beyond (i.e. to the sister genus *Procryptocerus*): 1) strong conservation of the microbial community composition across the genus; 2) conservation of N-recycling capacities and assignment of N-recycling genes to the same core symbiont taxa across all or nearly all metagenomes for the 16 studied species; 3) conservation of in vitro N-recycling capacities for symbionts from multiple *Cephalotes* species; and 4) conserved functions inferred from the genomes of symbionts isolated from distantly related host *Cephalotes* species.

We have attempted to make our argument clearer in the manuscript, as detailed in a response to one of this reviewer’s earlier comments, and hope that the approach now adequately spells out where evidence comes from just one vs. multiple species.

Line 139: Two of these samples

>>We have changed to “Two of these libraries”

Line 144: What do you mean here by host-specific clades? Do you mean that the taxonomy of the communities was conserved across Cephalotes, but that each of the Cephalotes harbored different OTUs/genera/etc? I think a few additional details about the results of this previous study would be useful and help with the interpretation of the results from the metagenomic sequencing.

>>We agree that this is a useful clarification. This specific use of “host-specific clades” is now clarified with the clause, “i.e. lineages of bacteria distributed across Cephalotes and its sister genus, Procryptocerus, that are, thus far, found only in these hosts⁴³.”

Line 145: Blast analysis of the 200 bp fragments is not the best method for taxonomically classifying 16S sequences. For instance, taxonomic informativeness varies greatly across 16S rRNA. Sequence data from some regions may only bin to higher taxonomic levels while other regions carry more family and genus specific sequence signatures. What measures were taken to control for this variation? What databases were searched? NT? Other? If NT, why wasn't a dedicated 16S database used for taxonomic classification (RDP)? Such databases are often more accurately curated than the sequences housed in NCBI and can carry more reliable taxonomic information. Lastly, what was the % similarity to these previously identified taxa? Figure 3. I am having a hard time finding the N-recycling bacteria in the tree. Consider increasing the weight on those branches to make them stand out more.

>>We have updated the text to clarify that BLAST searches were not performed for taxonomic analysis of these 16S regions (those classifications were based on IMG-based scaffold classification, which was not reliant only 16S rRNA, but on any other sequences from scaffolds encoding 16S rRNA genes). BLAST searches were used to recover sequences for our 16S rRNA phylogeny, which aided in our identification of taxa from cephalotine-specific lineages. To clarify our use of BLAST, this passage now reads, “To obtain the closest available known relatives, top BLASTn hits against the NCBI non-redundant nucleotide database were downloaded for each sequence, and jointly used in a maximum likelihood phylogenetic analysis.” Later on in our responses to Reviewer #3 we address questions of % sequence identity, but we note here that metagenome-derived 16S rRNA sequences from the present study (i.e. those from core/specialized lineages) showed high % identity (sometimes 100%) to sequences in the GenBank database and to sequences from cultured symbiont isolates. This is partially illustrated, by short branch length separation, in Fig. S5. In Fig. S15, we use a 16S rRNA-independent approach to show how the genomes of cultured isolates were highly similar to those of strains found in abundance within our metagenomes.

Line 155: At what taxonomic rank the microbiome conserved? Class level? Or at finer taxonomic levels (genus, etc)? The authors should clarify those findings in the text.

>>We have clarified the sentence as follows: “Results from these analyses paralleled prior 16S rRNA-based discoveries of a highly conserved core microbiome, in which different *Cephalotes* species host microbes from a limited set of lineages found exclusively across this genus and the sister genus, *Procryptocerus* (Figs. 3, S5). These symbiont lineages nest within described orders from the Proteobacteria, Verrucomicrobia, and Bacteroidetes. While several of these bacteria await formal taxonomic descriptions, work on two core lineages shows that some comprise novel genera⁴⁴ and, in at least one case, a novel family⁴⁵”

Line 156: The authors mention that *Opiritales* was the most dominant order detected in all of the samples. However, in Figure S1 (16S amplicon analysis), that does not always appear to be the case in the three colonies presented in the amplicon analysis. This should be addressed.

>>Based on this comment we have updated the legend of Figure S1 to point out why the results might differ (see below). At the same time, we are comfortable with the approach in the main manuscript on line 156. In that section we are only trying to point out that *Opiritales* were the most abundantly sampled symbionts in our shotgun metagenomics sequencing (method 1, targeting dissected mid- and hind-guts). This was true for 17 out of 18 metagenome libraries. Diving into results of Figure S1 (method 2, targeting gasters with Illumina amplicon sequencing of the V4 region of 16S rRNA), and how/whether these say something different about abundance, gets complicated, based on interpretations from previous publications, and using data from a manuscript we have in preparation.

For instance, sequencing of 16S rRNA genes using 454 amplicon pyrosequencing of 16S rRNA (method 3, targeting dissected mid- and hind-guts; Hu et al. 2014, Kautz et al. 2012), or Sanger sequencing of PCR-amplified and cloned 16S rRNA genes (method 4, targeting dissected mid- and hind-guts; Russell et al. 2009, Kautz et al. 2012) often place *Opiritales* as the dominant symbiont within these gut chambers—resembling the results shown for Figure 3. Yet, like the results shown in Figure S1 (focused on gasters), Illumina sequencing of the V4 regions targeting whole adult workers suggest a lower relative abundance for the *Opiritales* symbiont group (method 2, focusing on whole adult workers; Hu et al. manuscript in preparation). Bacterial symbionts of *Cephalotes* can be found in the crop (i.e. not part of our mid- and hind-gut preparations; but part of our gaster preparations) and infrabuccal pocket (found in the head, so present only when sampling whole ant workers). So there are places besides the mid-gut and hind-gut where microbes reside. Since: 1) multiple studies focused on mid- and hind-guts often point to *Opiritales* as the most abundant symbionts summed across these chambers, and 2) studies arguing against *Opiritales* dominance have involved the use of different sequencing methods or targeting of additional tissues, it seems complicated to be wading into a discussion of why Figure S1 might differ from our findings in Figure 3.

Since the manuscript does not focus on establishing symbiont abundance, we would hence prefer to keep this section as is to aid in keeping the narrative focused. We hope this seems reasonable to this reviewer and the editor.

Given the reviewer's observation, however, we do propose to add the following to the legend of Figure S1: "DNA samples extracted from gasters of worker ants at the end of our 15/14N glutamate experiments were used for 16S rRNA amplicon sequencing (those from other experiments were not subjected to sequencing). This approach differs from that used for DNA preparation in our shotgun metagenomics sequencing, which targeted dissected mid- and hind-gut tissue. Differing protocols, or perhaps taxon bias in the differing molecular methods, could account for differences seen in symbiont relative abundance in this figure and Figure 3."

Line 159: co-existence of distinct strains...the 10% difference in GC content suggests that the different Xanthomonadales strains that the authors refer to may belong to different genera or families. Suggest using the term 'operational taxonomic unit' (OTU) instead of strain here for that reason. Same comment for line 162 and also throughout the manuscript. Also, it is difficult to read the axes labels on figure 3, so it is hard to determine how different the depths of coverage are for the read clouds. Small differences in depth may not indicate different strains and, likewise, genomic regions can vary in terms of GC content. Being able to more clearly see the level of separation would greatly aid in the assessing the author's claim that these different read clouds indicate different OTUs. Do the rRNAs pulled from the assembly also support multiple OTUs?

>>We agree that the large difference in GC content suggests larger-than-strain-level variation in Xanthomonadales taxon bins, and have changed this to "lineages" from "strains." We would prefer "lineages" over "OTUs" here, as we are not picking an operational definition—rather, we are using an established taxonomic threshold (order).

In other parts of the manuscript we have retained the usage of strain, however, as the similar GC content, distribution, and poor assembly of these bins are all consistent with strain-level variation as it is typically used in metagenomics studies.

We note that 16S rRNA based mining from metagenomes does indeed suggest the presence of multiple strains from particular taxonomic groups. In our best sampled metagenome, PL010 from *C. varians*, there were 29 different 16S rRNA copies from Burkholderiales (granted some of these may be non-overlapping portions of the same 16S rRNA from a single strain), 5 different Rhizobiales 16S rRNA copies, and 3 different Xanthomonadales 16S rRNA copies. In contrast, we found just one 16S rRNA copy from the Opitutales, Pseudomonadales, Campylobacteriales, Flavobacteriales, and Sphingobacteriales. The higher diversity of the former three orders vs. the latter five is reflected in the apparent differences in numbers of distinct scaffold clouds for this colony's metagenome library in Figure 3. Findings of 16S rRNA copy number from other metagenomes are consistent with this overall trend (i.e. 7-14 16S rRNA copies for Burkholderiales; 1-3 copies for Rhizobiales; 1-6 copies for Xanthomonadales; 1 copy for Opitutales; 1-2 copies for Pseudomonadales; 1 copy for Campylobacteriales; 1 copy for Sphingobacteriales, and 1-3 copies for Flavobacteriales). And evidence for the presence of multiple strains per order has been provided additionally in prior study (e.g. Hu et al. 2014).

To address concerns over difficulty in reading Figure 3 we have increased font size on the x- and y-axes, and we thank the reviewer for pointing out this area for improvement.

Line 161: though still ubiquitous across metagenomes,

>>We have added the suggested comma.

Line 163: I think you can change 'each taxon' to 'each order' for more clarity.

>>We agree, and have changed accordingly.

Line 165: It is not clear how presence/absence calls for N metabolism genes suggested rarity of these symbionts? A better indicator would be abundance of 16s rRNA or another universally conserved gene.

>>We agree, and have added a citation to a more specific taxonomic survey across the genus. We have also reworded the section regarding N-metabolism genes to clarify that the point of noting these genes is not really to establish the abundance or distribution of the organisms, but to explain how the observed distribution and abundance of these organisms is likely to affect the observed pattern of N-metabolism genes. “were common but not ubiquitous, echoing previous results using 16S rRNA amplicon sequencing^{35,43}. The absence or rarity of these symbionts in several metagenomes is also reflected in the presence/absence calls for N-metabolism genes in these taxa (Table S5).”

Lines 173-176: Absence of nitrogenase genes in the metagenomic assemblies from insects that were not tested for acetylene reduction could simply be due to low abundance. Thus, the authors should not completely exclude the presence of nitrogen fixing bacteria unless more specific PCR-based analysis targeting nifH were also employed (and were also negative).

>>We agree that rare members of the community could encode nitrogenase genes, and would argue that our current wording on lines 174-176 accounts for this possibility: “Together, our experiments and metagenomics suggest that prior observations of nifH genes in Cephalotes workers arose from detection of rare or contaminant bacteria¹⁷ or from bacteria colonizing a portion of the gut not included not targeted in our metagenomics study.”

Line 192: change additional bacteria encoding complete sets to additional complete sets

>>The passage “hosted additional bacteria encoding complete sets of urease core and accessory proteins” was meant to point out that a second (or third) symbiont encoding this function was present. The language in the reviewers’ suggested wording, i.e. “hosted additional complete sets of urease core and accessory proteins” removes the focus from the intended subject (i.e. other symbionts encode urease function) and takes an unusual approach of saying that ants host proteins, rather than saying that they host symbionts encoding the specific proteins. For these reasons, we prefer something closer to the original wording, with a slight tweak” “hosted additional bacteria encoding the core and accessory proteins required for urease function”

Lines 213-214: I think this sentence is missing a word or two. Please check. Also, typo...extensive. Additionally, it is still not clear what the authors mean by ‘core taxa.’ Do you mean core orders (as in genes involved in these processes were assigned to bacterial orders found in association with all Cephalotes)? Please make sure this term is very clearly defined early on in the results section.

>>We have fixed the identified typo (i.e. “extensiove”). To clarify, we have changed this sentence to “Metagenome-derived predictions of functional capacity met these expectations in *C. varians* and all 16 other turtle ants.”

With regard to the vagueness of the term “taxa”, we refer to our response to an earlier comment by this reviewer. To re-iterate, some of these symbiont lineages have been assigned to, or will likely correspond to, novel genera. Others have been assigned to, or will likely assign to, novel families. Without the formal taxonomic descriptions for all symbiont groups it is difficult to pin down the unit we should be referring to. Since we have now explicitly defined ‘core taxa’ earlier on in the Results section (based on this reviewer’s comments), we prefer to stick with this term.

Line 222: It is unclear what the authors are referring to here....’This paralleled.’ What does ‘this’ refer to? Please revise.

>>Per a comment from another reviewer, this whole section has been shortened in revision, and no longer contains this phrase.

Line 235: The authors mention that several genes related to uricase activity as well as one canonical uricase gene were binned to Hymenoptera. Is it possible that the insect can perform these functions on its own without the help of bacteria? It would be interesting to perform a uricase assay on antibiotic treated and controls to see if uricase function declines in treated individuals. I have also noticed that large numbers of insect genomes (Hymenoptera and Coleoptera) contain genes related to uricase function and have wondered if these paths are functional.

>>This is an interesting question. We have made a revision earlier in the text touching on this; please see our response to ‘major point 3’ above. While our future research will certainly keep in mind the hosts’ contribution to N-waste metabolism, we would argue that it is beyond the scope of the present manuscript.

Line 256: as genes for guanine deaminase enzymes were identified...

>>We have made this more explicit: “Core symbionts appear to produce xanthine via purine recycling: genes for guanine deaminase enzymes (Fig. S13) were classified to Burkholderiales bins in 16 metagenomes (Table S5).”

Line 266: metagenomes, but only....

>>We thank the reviewer for the suggestion. We have changed to “Genes for a separate, two-step pathway converting arginine to urea (Fig. S14) were present in most metagenomes; only Burkholderiales encoded both steps.”

Line 270: Again, I would choose another word besides strains as that implies more than one strain from an individual bacterial species. I would use OTUs instead.

>>As noted above, we disagree with the use of “OTU” in this context. Here, we have substituted “isolates” where specifically referring to the individual cultured isolates sequenced, and retained “strains” where used in contexts for which there is substantive reason to believe that the commonly-held interpretation of ‘strain’ applies.

Line 273: We addressed this shortcoming

>>We have changed to “We addressed this limitation through genome sequencing of...”

Lines 273-275: It is not completely clear how the bacterial isolates were selected for full genome sequencing. More explanation is needed here. For example, five of the eight dominant ‘taxa’...what does that mean? Dominant orders? Species? Etc? Were the 14 strains selected from these ‘taxa’? The authors mention that these core bacteria were shared among two of the ant species analyzed? Were there any core/conserved bacteria/OTUs shared among all of the ant species? Or alternatively, were there few common OTUs? Either observation would be interesting

to point out.

>>To clarify the vagueness of “core taxa”, and to more clearly emphasize that these symbionts were major players in gut communities of targeted hosts, we have reworked the passage as follows: “We addressed this limitation through genome sequencing for 14 cultured symbionts isolated from two distantly related *Cephalotes* species, *C. varians* and *C. rohweri*. These targeted isolates hailed from widely distributed, cephalotine-specific lineages that spanned five of the eight orders containing core symbiont taxa. Furthermore, they were identical or nearly identical at 16S rRNA gene sequences in comparison to abundant symbiont strains sampled using culture-independent methods (Fig. S5).”

We hope that the updated phrasing addresses questions of whether these came from widely distributed OTUs—the answer to this question was indeed yes. Going into greater detail, however, gets complicated. For instance, within several host specific lineages (e.g. those in the Burkholderiales) there are often sub-lineages, each showing widespread distributions across multiple *Cephalotes* species. The apparently complicated history of host switching, co-diversification, symbiont loss, lineage duplication, etc. requires much more sampling and more sophisticated phylogenetics (i.e. not just 16S RNA). Until that work has been done, it is difficult to provide the exact details requested here by the reviewer. We hope the revised approach strikes the right balance.

Line 277: Were the 16s rRNA genes from the draft genomes also similar to 16s genes in the metagenome assembly? That would lend further proof that the strains selected for sequencing are indeed present in vivo.

>>As stated above, the 16S rRNA genes from cultured isolates were highly similar to those obtained from previously published studies. They were also highly similar and sometimes identical to those derived from the present shotgun metagenomics efforts. For illustration in Figure S5, see the minimal branch lengths separating cultured isolates’ 16S rRNA sequences (with dark red highlighting) vs. metagenome-derived 16S rRNA sequences (in pink shading; but also note that metagenome-derived 16S rRNA sequences have names like “species name_scaffold identifier”). For instance, the sequence “VARIANSPL005W_12103201” from the metagenome of *C. varians* colony PL005 was identical to the 16S rRNA of cultured isolate Cv44, also derived from *C. varians*.

Please note that we have also aligned metagenome-derived scaffolds to scaffolds from genome assemblies. Because of the slow rate of 16S rRNA gene evolution, this additional approach gives a greater ability to detect close relatedness. Our findings indeed show that cultured isolates share high relatedness to strains found in abundance via metagenomics (Fig. S15).

Line 283: This sentence is confusing. The strain lacked urease genes but three cultured symbionts encoded all genes necessary for urease function....do you mean three other cultured symbionts coded for urease and all of the necessary accessory genes?

>>We agree and thank the reviewer for their suggestion. We have clarified as follows: “While this isolate lacked urease genes, suggesting an inability to convert the derived urea into ammonia, three other cultured symbionts encoded all genes necessary for urease function, including *Cephaloticoccus* isolates from *C. varians* (Cv41) and *C. rohweri* (Cag34) and a Xanthomonadales symbiont from *C. rohweri* (Cag60).”

Line 295: What symbiont core taxa are you referring to here?

>>Thanks for this comment. We have changed the sentence to “The 11 near complete draft genomes, where >87% of universal single copy genes were detected, spanned seven of the eight orders containing core symbiont lineages, i.e. all but Pseudomonadales (Fig. 5; Tables S8, S9, S10).”

Line 297: How were the dominant strains identified? By coverage?

>>Yes, these strains were deemed as being dominant because they had sufficient sequence coverage to allow >87% of their single copy genes to be detected. Table S9 shows the depths of sequence coverage for scaffolds assigning to each bin. Here one can see that average read depth for each scaffold, for these bins, ranged from 14.71 fold to 1966 fold coverage.

Line 307: expanding the number of targeted cephalotine species...suggesting revising this sentence to, expanding the number of targeted cephalotine-associated bacterial species. The way it is worded now makes it sound like you preformed urease assays on the ants themselves (similar to the N15-urea experiments).

>>Thanks to the reviewer for this suggestion. We have changed this to “we performed a series of in vitro assays from isolates cultured from an expanded number of cephalotine species.”

Line 328: I would remove the phrase large amounts because the boosts in essential amino acid levels observed by the authors were small under the experimental conditions. It is possible that other nutritional sources could provide amino acids, but I agree that the bacteria are likely an important component of N acquisition and retention in this system.

>>We disagree that the observed levels of bacterial-derived N were small—we found 15-36% of essential amino acids in our experimental system showed incorporation of N from labeled urea, an elevation that went away when bacteria were suppressed with antibiotics. On lines 337-344 we discuss similar attempts to estimate the amounts of symbiont-provisioned N in other systems. These quantities, obtained with just 5 weeks of feeding by adults, suggest only slightly lower symbiont contributions in the present system vs. those seen in systems where symbionts are clearly making

major contributions. We would also point out the large differences in the incorporation of dietary N from urea versus that seen for the N or C from dietary glutamate (Figs. S2 & S3).

Line 355: has thus emerged...I don't think you need the commas surrounding thus.

>>Thank you for catching this; commas have been removed.

Line 356: has evolved separately in association with unrelated, yet functionally similar symbionts. Separately within what context? Herbivorous ants in general or within the cephalotine ants? An additional detail or two will clarify.

>>We have added some additional clarifying details: “—Blochmannia in the Camponotini, and the diverse bacteria described here in the Cephalotini.”

Line 434: How do you know these scaffold assignments are incorrect? Could these microbes have been consumed with food and be present in the gut? More details would be useful.

>>We thank the reviewer for this suggestion and provide more information about “incorrect scaffold assignments”, “to errant bacterial taxa”. Here is the revised passage “Scaffolds were occasionally binned to taxa not predicted to be found in the Cephalotes gut microbiome. For example, urease-gene-encoding scaffolds classified to the order Rhodocyclales. Prior 16S rRNA based studies using various universal primer pairs had failed to detect Rhodocyclales symbionts in Cephalotes ants. Since these particular scaffolds showed similar depths of coverage to scaffolds from Opitutales, and since the encoded urease genes were identical to those found in cultured Opitutales (i.e. Cephaloticoccus), we concluded that this result reflected a classification error. To correct for this, and to query other scaffolds with unexpected classifications, we compared such scaffolds to reference genomes from seven cultured bacterial isolates, hailing from core Cephalotes-specific lineages.”

794: It is difficult to see the circle sizes in Figure 3 (taxonomy) to indicate the scaffold sizes for each lineage. Also, are these N50 scaffold sizes or total scaffold lengths?

>>These are meant only to give an indication of scaffold size. They are per-scaffold—N50 would make sense only in the context of a collection of contigs. We have changed wording in caption to “Circle size indicates length of each scaffold.” While we recognize that it will be hard to differentiate scaffolds at the smallest size classes at normal magnification, we note that readers viewing the manuscript on a computer will have the option of zooming in to do so. Furthermore, the largest scaffolds are easy to differentiate from the smallest ones.

Lines 818-820: It is unclear how the authors are suggesting that there is likely only one *Opitutales* strain. Please provide more information about why this is likely.

>>We have updated with a parenthetical to indicate why this is likely: “(suggested by large contigs with similar depth of coverage in our metagenomes, Fig. 3, and by the presence of just one *Opitutales*-assigned 16S rRNA gene from each metagenome, Fig. S5)”

Figure 6: The box colors for the corresponding taxonomic assignments should be defined either on the figure or in the legend.

>>Good suggestion. We have now included the requested assignments at the bottom of the figure.

Reviewer #4 - Expertise: Isotope ecology & labeling experiments

This study tested the hypothesis that *Cephalotes* ants exploit organic nitrogen derived from their symbiotic microbes. I have a few comments on the manuscript as follows.

L112: Why was the effect size of labelled glutamate small? As illustrated in Fig. 4, Glu is the anchor compound bridging between ammonia and amino acids. I would expect the same results with the urea-feeding experiment. However, Fig. S4 indicates that all amino acids with the exception of Phe were not derived from ¹⁵N-labelled Glu. This may suggest that the illustration in Fig. 4 is too much simplified

>>This is a good question. Since most nutrient absorption in insects occurs within the midgut, one explanation is that glutamate crossed the gut wall prior to symbionts having access to it (i.e. symbionts in this system appear most abundant at the anterior part of the midgut and further downstream in the ileum, based on prior electron microscopy studies; e.g. Roche & Wheeler 1997; [http://onlinelibrary.wiley.com/doi/10.1002/\(SICI\)1097-4687\(199712\)234:3%3C253::AID-JMOR4%3E3.0.CO;2-A/full](http://onlinelibrary.wiley.com/doi/10.1002/(SICI)1097-4687(199712)234:3%3C253::AID-JMOR4%3E3.0.CO;2-A/full)). We also note that most symbionts can derive glutamate internally, i.e. via NH₃ assimilation. Hence, our results could still be explained if glutamate reached symbionts in the posterior midgut and ileum, if the symbionts lacked the transporters for glutamate uptake or if conditions were such that they invested little in this process. It is further possible that symbiont metabolism is most active in the presence of waste nitrogen or when there are limiting quantities of amino acids in the diet. It may, thus, be relevant that the urea experiments included no other sources of nitrogen, while glutamate experiments included non-essential amino acids (and no N-waste).

Due to the aforementioned explanations, and the genome data, we don't see any reasons to deviate from the model proposed in Figure 4. But we fully agree that it is important to point out that this is a hypothesized model, and one that is need of empirical testing. Please see below for the passage we

have added to the end of the Results section, which makes an effort to address the above concerns as well as those raised for the content on Line 128. We thank the reviewer for providing this critique, as we feel that the manuscript has improved by making the details and uncertainties of this model more explicit.

L128: Why did only asparagine in the heavy urea treatment show insignificant ¹⁵N percentage with the controlled treatment, although it is primarily derived from aspartic acid that showed a significant difference? A brief but explicit explanation may be needed here

>>This is another good question. While there is no way to know from these results alone, these figures, which indicate circulating amino acids in the host hemolymph, will reflect the combined effects of metabolic capacity of the host+microbiome consortium and dynamics of amino acid turnover. Although not significantly different in labeled Asn from the bacteria-replete treatment group, both labeled groups showed higher median incorporation of ¹⁵N in Asn than the unlabeled control (Fig. 2). This, combined with the fact that most non-essential amino acids in the bacteria-reduced (i.e. antibiotic) treatment showed less ¹⁵N incorporation relative to the unlabeled control, suggests that 1) there was still incorporation in antibiotic-treated ants but at a potentially lower rate, and that 2) the lower overall rate of incorporation of urea-derived N may have still been enough to keep up with lower turnover in Asn compared to the other amino acids.

Another consideration is that we do not yet know how these amino acids are acquired by hosts. It could be that they digest symbiont cells. It could also be that symbionts export them. Selective export of certain amino acids by bacteria or export by a subset of the symbionts (which may themselves be unable to make all amino acids), could plausibly explain these findings.

We allude to this question in the below passage, which has been placed at the end of the Results section. This is also intended to address the reviewer's above comment pertaining to content on Line 112:

“An efficient N-economy in *Cephalotes holobionts*: symbiont roles and pathways for N-flow

Our proposed model for symbiont contributions toward the *Cephalotes* N-economy is presented at the top of Fig. 4. In short, we predict that waste N is acquired by symbionts through a combination of ant metabolism, diet, and symbiont metabolism. Such N is recycled by a limited range of bacteria, prior to being assimilated into amino acids by most members of the microbiome. The majority of abundant symbionts can make most essential amino acids, and it is posited that N-entry from waste products into glutamate serves as a gateway for N-transfer to the amino acid pool, given glutamate's role as an NH₃ donor in numerous transamination reactions.

Numerous aspects of this hypothesized model await testing through direct experimentation. Supporting this model will further require an explanation as to why N from dietary glutamate could have a small relative impact on host amino acid pools (Fig. S4), if glutamate-facilitated transamination is major port of entry for symbiont-recycled N into the amino acid pool. Possible explanations

include absorption of glutamate across the midgut wall before it can reach symbionts, which are concentrated in the posterior midgut and throughout the ileum²². It may also be possible that a large influx of waste N (e.g. dietary urea) can activate symbiont metabolism to a greater degree than an influx of non-essential dietary amino acids, such as glutamate. In addition, the capacities for symbionts to derive glutamate internally (i.e. via ammonia assimilation), as inferred from (meta)genomes, raise questions about the necessity of glutamate import by Cephalotes' gut symbionts. Future experiments on the identities of nutrients imported and exported by gut symbionts will be informative. Similar efforts should elucidate whether hosts acquire symbiont-derived N by absorption of symbiont-exported amino acids across the gut, or through absorption of N acquired through the digestion of symbiont cells."

Fig. 1B: In y axis, delta 15N is not necessarily analogous with trophic level. In theory, it should be calculated from an offset against a given baseline. See Post (2002) Ecology 83:703-718 for more details. Furthermore, Cephalotes ants are no longer "herbivore" and their trophic level should be one unit higher than that of symbiotic bacteria. The authors may be interested in Steffan et al. (2015) PNAS 112:15119-15124

>>We agree that delta 15N is not exactly analogous with trophic level, and have modified the axis label of Fig. 1B accordingly. Thanks to the reviewer for this suggestion. We have altered the figure legend, accordingly: "(B) Nitrogen isotope data from prior studies^{14,31} targeting several New World locales. Included on each graph are isotope ratios for Cephalotes ants, other ants in their subfamily (Myrmicinae), and Camponotus ants, which host known N-recycling bacteria. Also plotted are data for plants, insect herbivores, and insect carnivores from the same locales. Isotope fractionation results in a gradual increase in the relative amounts of heavy (¹⁵N) versus light (¹⁴N) nitrogen, as one moves up the food chain. As such, ants with low amounts of the heavy nitrogen isotope, shown here using the $\delta^{15}\text{N}$ statistic, are argued to feed at low trophic levels."

Given uncertainty about the proper baseline in this case, and because the primary point we hope to convey with this panel is the relative placement of different organisms from each location, we believe it is more appropriate to retain the raw delta 15N numbers rather than presenting baseline-adjusted figures.

Regarding expected ¹⁵N offsets between Cephalotes and their symbiotic bacteria, we respectfully disagree with the argument that a simple "one additional trophic level" is a proper analogy. The systems characterized by Steffan et al (2015) may not be good analogues of internal microbiota with respect to expected 15N trophic offsets—high rates of nitrogen recycling could very well decrease the endpoint fractionation observed for the system, if much of the light nitrogen that is fractionated during catabolism is recaptured and reincorporated into host tissue. (Indeed, in the ecosystems surveyed here, sap-feeding herbivores are frequently less enriched in 15N than leaf-chewing herbivores.)

REVIEWERS' COMMENTS:

Reviewer #3 (Remarks to the Author):

The authors have done a thorough job addressing comments from each of the four reviewers. The incorporated edits have greatly strengthened the argument for the involvement of urea recycling in nitrogen acquisition and retention in turtle ants and the results and their interpretations are much clearer for those readers who may not be as familiar with the field of metagenomics. Overall, this manuscript makes significant contributions to the understanding of host-symbiont interactions and the conservation of this interaction across multiple ant species will certainly be of great interest.

I noticed one typo on line 119, but otherwise, I have no further comments.

to synthesize the isoleucine...

Reviewer #4 (Remarks to the Author):

I carefully examined the authors' responses to my comments as well as the revised manuscript. It seems to be fine to me. The authors may use a superscript for delta ¹⁵N on y axis of Fig. 1B.

Comments made by two reviewers from our second submission

Reviewer #3 (Remarks to the Author):

The authors have done a thorough job addressing comments from each of the four reviewers. The incorporated edits have greatly strengthened the argument for the involvement of urea recycling in nitrogen acquisition and retention in turtle ants and the results and their interpretations are much clearer for those readers who may not be as familiar with the field of metagenomics. Overall, this manuscript makes significant contributions to the understanding of host-symbiont interactions and the conservation of this interaction across multiple ant species will certainly be of great interest.

I noticed one typo on line 119, but otherwise, I have no further comments.

to synthesize the isoleucine...

>>We greatly appreciate the reviewer's positive comments. We have corrected the typo error as suggested by the reviewer #3.

Reviewer #4 (Remarks to the Author):

I carefully examined the authors' responses to my comments as well as the revised manuscript. It seems to be fine to me. The authors may use a superscript for delta "¹⁵" N on y axis of Fig. 1B.

>>We thank the reviewer for taking time to review our manuscript again and appreciate the positive assessment this time. We have changed "15" on y axis to a superscript in Figure 1B.